# Unique-region phosphorylation targets LynA for rapid degradation, tuning its expression and signaling in myeloid cells

Ben F Brian IV[1], Adrienne S Jolicoeur[1], Candace R Guerrero[2], Myra G Nunez[1], Zoi E Sychev[1], Siv A Hegre[3], Pål Sætrom[3,4], Nagy Habib[5], Justin M Drake[1,6,7], Kathryn L Schwertfeger[1,6,8,9], Tanya S Freedman[1,6,8,10]*

[1]Department of Pharmacology, University of Minnesota, Minneapolis, United States; [2]College of Biological Sciences Center for Mass Spectrometry and Proteomics, University of Minnesota, Minneapolis, United States; [3]Department of Clinical and Molecular Medicine, Norwegian University of Science and Technology, Trondheim, Norway; [4]Department of Computer Science, Norwegian University of Science and Technology, Trondheim, Norway; [5]Department of Surgery and Cancer, Hammersmith Hospital, Imperial College London, London, United Kingdom; [6]Masonic Cancer Center, University of Minnesota, Minneapolis, United States; [7]Department of Urology, University of Minnesota, Minneapolis, United States; [8]Center for Immunology, University of Minnesota, Minneapolis, United States; [9]Department of Laboratory Medicine and Pathology, University of Minnesota, Minneapolis, United States; [10]Center for Autoimmune Diseases Research, University of Minnesota, Minneapolis, United States

*For correspondence:
tfreedma@umn.edu

Competing interests: The authors declare that no competing interests exist.

**Abstract** The activity of Src-family kinases (SFKs), which phosphorylate immunoreceptor tyrosine-based activation motifs (ITAMs), is a critical factor regulating myeloid-cell activation. We reported previously that the SFK LynA is uniquely susceptible to rapid ubiquitin-mediated degradation in macrophages, functioning as a rheostat regulating signaling (Freedman et al., 2015). We now report the mechanism by which LynA is preferentially targeted for degradation and how cell specificity is built into the LynA rheostat. Using genetic, biochemical, and quantitative phosphopeptide analyses, we found that the E3 ubiquitin ligase c-Cbl preferentially targets LynA via a phosphorylated tyrosine (Y32) in its unique region. This distinct mode of c-Cbl recognition depresses steady-state expression of LynA in macrophages derived from mice. Mast cells, however, express little c-Cbl and have correspondingly high LynA. Upon activation, mast-cell LynA is not rapidly degraded, and SFK-mediated signaling is amplified relative to macrophages. Cell-specific c-Cbl expression thus builds cell specificity into the LynA checkpoint.
DOI: https://doi.org/10.7554/eLife.46043.001

## Introduction

Phosphorylation of immunoreceptor tyrosine-based activation motifs (ITAMs) by Src-family kinases (SFKs) is the first enzymatic step in the activation of an innate immune response during a pathogen encounter. Initiation of cell-activating signaling typically occurs within clusters of ITAM-coupled receptors such as FcγR (*Guilliams et al., 2014*) or the hemi-ITAM Dectin-1 (*Brown, 2006*; *Deng et al., 2015*), nucleated by highly multivalent immunoglobulin-G-decorated pathogens or fungal cell-wall β-glucans, respectively. Together, SFKs and phosphorylated ITAMs trigger activation of the tyrosine kinase Syk (*Lowell, 2011*). The SFKs and Syk then drive activation of membrane-

proximal signaling through adaptor proteins; cytoskeleton-modulating proteins, such as the FAK/Pyk2 kinases and Paxillin; and Tec kinases, which activate second-messenger pathways via phosphoinositide 3-kinase (PI3K) and phospholipases Cγ (PLCγ1/2). These early events ultimately lead to downstream signaling through Erk1/2 and other pathways (*Lowell, 2011*). As the byproducts of activated macrophages can be toxic (*e.g.* release of reactive oxygen species) and drive inflammation (*e.g.* release of tumor necrosis factor α), the responsiveness of innate immune cells is tightly regulated (*Goodridge et al., 2011*; *Takai, 2002*; *Sondermann, 2016*; *Chiffoleau, 2018*).

Multiple mechanisms work together to tune the responsiveness of macrophages and other myeloid cells, including negative regulation by the phosphatases CD45 and CD148 (*Goodridge et al., 2011*; *Freeman et al., 2016*; *Bakalar et al., 2018*), cytoskeletal barriers to diffusion (*Jaumouillé et al., 2014*), signaling via immunoreceptor tyrosine inhibitory motifs (ITIMs) (*Abram and Lowell, 2008*) and inhibitory ITAMs (*Hamerman and Lanier, 2006*; *Hamerman et al., 2009*), and degradation and sequestration of signaling molecules targeted for polyubiquitination by ubiquitin ligases (*Lutz-Nicoladoni et al., 2015*; *Liyasova et al., 2015*). The SFKs, which in myeloid cells typically include Fgr, Fyn, two isoforms of Hck, and two splice forms of Lyn, may also have positive and negative functions (*Abram and Lowell, 2008*; *Mkaddem et al., 2017*; *Alvarez-Errico et al., 2010*; *Scapini et al., 2009*). Layered onto the traditional positive- and negative-regulatory roles of the SFKs, activated LynA (the longer of the two Lyn splice forms) is rapidly and specifically targeted for polyubiquitination and degradation, forming the basis of a signaling checkpoint that blocks spurious macrophage activation (*Freedman et al., 2015*). This checkpoint can be bypassed when LynA is upregulated, with LynA acting as a rheostat to tune macrophage sensitivity. For instance, LynA is transcriptionally upregulated when macrophages are treated with interferon (IFN)-γ, and these primed cells have a lower threshold for SFK-mediated signaling (*Freedman et al., 2015*). The molecular mechanism that selectively targets LynA for polyubiquitination has not been elucidated previously.

Unlike Dectin-1 and FcγR signaling pathways in macrophages, which are typically triggered in the context of pathogen-induced μm-scale clusters of receptors (*Goodridge et al., 2011*; *Freeman et al., 2016*; *Freedman et al., 2015*), mast-cell FcεR signaling has a low threshold for activation; small or even monovalent antigen-IgE complexes can induce a signaling response in the context of a cell-particle interaction (*Andrews et al., 2009*; *Felce et al., 2018*; *Carroll-Portillo et al., 2010*). Supporting this more permissive signaling function, the binding dynamics of Syk with FcεRI in mast cells is unaffected by the size of receptor aggregates (*Schwartz et al., 2017*). Like macrophages, mast cells express LynA, and the disparity in the receptor sensitivity of these two cell types has not previously been explained.

This paper describes the mechanism by which activated LynA is selectively targeted for rapid degradation, thereby tuning both its steady-state expression and its activation kinetics in a cell-specific manner. To reveal the requirements for LynA degradation, we synchronized receptor-independent SFK activation using the designer inhibitor 3-IB-PP1 (*Okuzumi et al., 2010*; *Okuzumi et al., 2009*; *Schoenborn et al., 2011*), which specifically inhibits a variant of the SFK-inhibitory kinase Csk (Csk$^{AS}$) (*Freedman et al., 2015*; *Schoenborn et al., 2011*; *Tan et al., 2014*). Csk is responsible for phosphorylating a key inhibitory tyrosine in the C-terminal tail of all the SFKs. Inhibiting Csk$^{AS}$ with 3-IB-PP1 disrupts the dynamic equilibrium between Csk and the phosphatases CD45 and CD148, which dephosphorylate the inhibitory tyrosine, leading to rapid and robust SFK activation (*Chow and Veillette, 1995*; *Zhu et al., 2011*). 3-IB-PP1-induced SFK activation leads to the phosphorylation and likely activation of the E3 ubiquitin ligase c-Cbl (*Feshchenko et al., 1998*; *Dou et al., 2012*; *Freedman et al., 2015*) and the preferential polyubiquitination and degradation of LynA in macrophages (*Freedman et al., 2015*), but it was not clear whether these two events were linked. Using knockout and overexpression models coupled with analyses of protein abundance and cell signaling, we now demonstrate that c-Cbl controls the steady-state expression of LynA and mediates its rapid degradation upon activation in macrophages. Quantitative targeted mass spectrometry shows that LynA is targeted by c-Cbl in response to phosphorylation of tyrosine 32 (Y32) within the LynA unique region, a process mediated by LynA, LynB or the shorter isoform of Hck. This recognition mode is distinct from the slower-phase action of c-Cbl and other E3 ligases on LynB and

the other SFKs (*Freedman et al., 2015*; *Sanjay et al., 2001*; *Sanjay et al., 2006*). Finally, we have discovered that the LynA checkpoint is cell-specific. In mast cells, which express very little c-Cbl but abundant Cbl-b, LynA is not rapidly degraded upon activation, a function that can be rescued by induced overexpression of c-Cbl. The differential regulation of c-Cbl and LynA may drive mast cells toward a lower activation threshold than macrophages.

## Results

### c-Cbl mediates steady-state and activation-induced degradation of LynA in macrophages

We reported previously that activated LynA is rapidly polyubiquitinated and degraded in Csk[AS] macrophages treated with the Csk[AS] inhibitor (SFK activator) 3-IB-PP1 (*Freedman et al., 2015*). To identify the E3 ubiquitin ligase that mediates this degradation, we tested the functions of c-Cbl and Cbl-b, known modulators of ITAM signaling in both adaptive and innate immune cells (*Abram and Lowell, 2008*; *Lutz-Nicoladoni et al., 2015*; *Tang et al., 2019*). We bred Csk[AS]Cbl[-/-] and Csk[AS]Cblb[-/-] mice from the existing strains Cbl[-/-] (*Rafiq et al., 2014*) and Cblb[-/-] (*Chiang et al., 2000*) and generated Csk[AS]-expressing, c-Cbl-deficient (Csk[AS]c-Cbl[KO]) and Csk[AS]-expressing, Cbl-b-deficient (Csk[AS]Cbl-b[KO]) bone-marrow-derived macrophages (BMDMs). (For clarity we refer to BMDM subtypes using protein nomenclature.) Immunoblots of whole-cell lysates show the loss of expression of c-Cbl and Cbl-b in Csk[AS]c-Cbl[KO] and Csk[AS]Cbl-b[KO] BMDMs, respectively, compared to Csk[AS] BMDMs (*Figure 1A–B*). Resting (unprimed) BMDMs were then treated with 3-IB-PP1 to induce activation of the SFKs (*Figure 1A*).

Knockout of c-Cbl profoundly impaired the degradation of activated LynA in the first few minutes of 3-IB-PP1 treatment, with LynA levels remaining at 100% after 1 min of exposure to 3-IB-PP1 in Csk[AS]c-Cbl[KO] BMDMs compared to 60% in Csk[AS] BMDMs (*Figure 1C*). Csk[AS]c-Cbl[KO] BMDMs also expressed 3-fold more LynA protein than Csk[AS] at steady state (*Figure 1D*). The impaired degradation and increased steady-state LynA expression together resulted in 6- to 8-fold elevations in LynA protein in the first minutes of 3-IB-PP1 treatment.

Although it has been reported previously that c-Cbl and Cbl-b have some redundant functions (*Naramura et al., 2002*; *Purev et al., 2009*), we were unable to detect any role for Cbl-b in suppressing the steady-state level or promoting the activation-induced degradation of LynA. Upon treatment for 1 min with 3-IB-PP1, LynA was 81% reduced in Csk[AS]Cbl-b[KO] compared to 60% in Csk[AS] BMDMs (*Figure 1C*). Although not statistically significant, steady-state expression of LynA was slightly depressed in Csk[AS]Cbl-b[KO] BMDMs (*Figure 1D*). This increase in LynA degradation may be explained by a compensatory upregulation of c-Cbl protein expression in Csk[AS]Cbl-b[KO] BMDMs (*Figure 1B*).

In the above experiments, activating Syk phosphorylation was used as a control for 3-IB-PP1-induced SFK signaling. Notably, Syk phosphorylation is enhanced in 3-IB-PP1-treated Csk[AS]c-Cbl[KO] BMDMs, consistent with the higher expression level and longer half-life of LynA protein (*Figure 1A*). Syk phosphorylation is also enhanced in 3-IB-PP1-treated Csk[AS]Cbl-b[KO] BMDMs, but this is likely due to the direct role of Cbl-b on Syk inhibition (*Sohn et al., 2003*). Analysis of activating phosphorylation of Erk1/2, a downstream signaling protein, is also complicated by the functions of the E3 ligases themselves. Erk phosphorylation in 3-IB-PP1-treated Csk[AS]c-Cbl[KO] BMDMs is impaired (*Figure 1—figure supplement 1*), likely due to the loss of the adaptor function of c-Cbl in PI3K signaling (*Ueno et al., 1998*; *Hunter et al., 1999*). In 3-IB-PP1-treated Csk[AS]Cbl-b[KO] BMDMs, Erk1/2 phosphorylation, like Syk phosphorylation, is enhanced, again likely due to the negative-regulatory role of Cbl-b for Syk and other upstream signaling intermediates.

To test whether the enhancement of LynA degradation in Csk[AS]Cbl-b[KO] BMDMs was in fact due to c-Cbl upregulation, we complemented our observations in the Cbl knockout mice with experiments using small interfering (si)RNAs to knock down the expression of c-Cbl and Cbl-b. Csk[AS] BMDMs were transfected with non-targeting control RNA (ctrl) or with siRNA constructs targeting Cbl or Cblb mRNA. Rested transfectants were then treated with 3-IB-PP1 to induce SFK activation. Immunoblots of siRNA-transfected BMDM lysates (*Figure 1—figure supplement 2A*) showed modest but specific knockdown of c-Cbl (70% reduced) and Cbl-b (60% reduced), importantly, without a

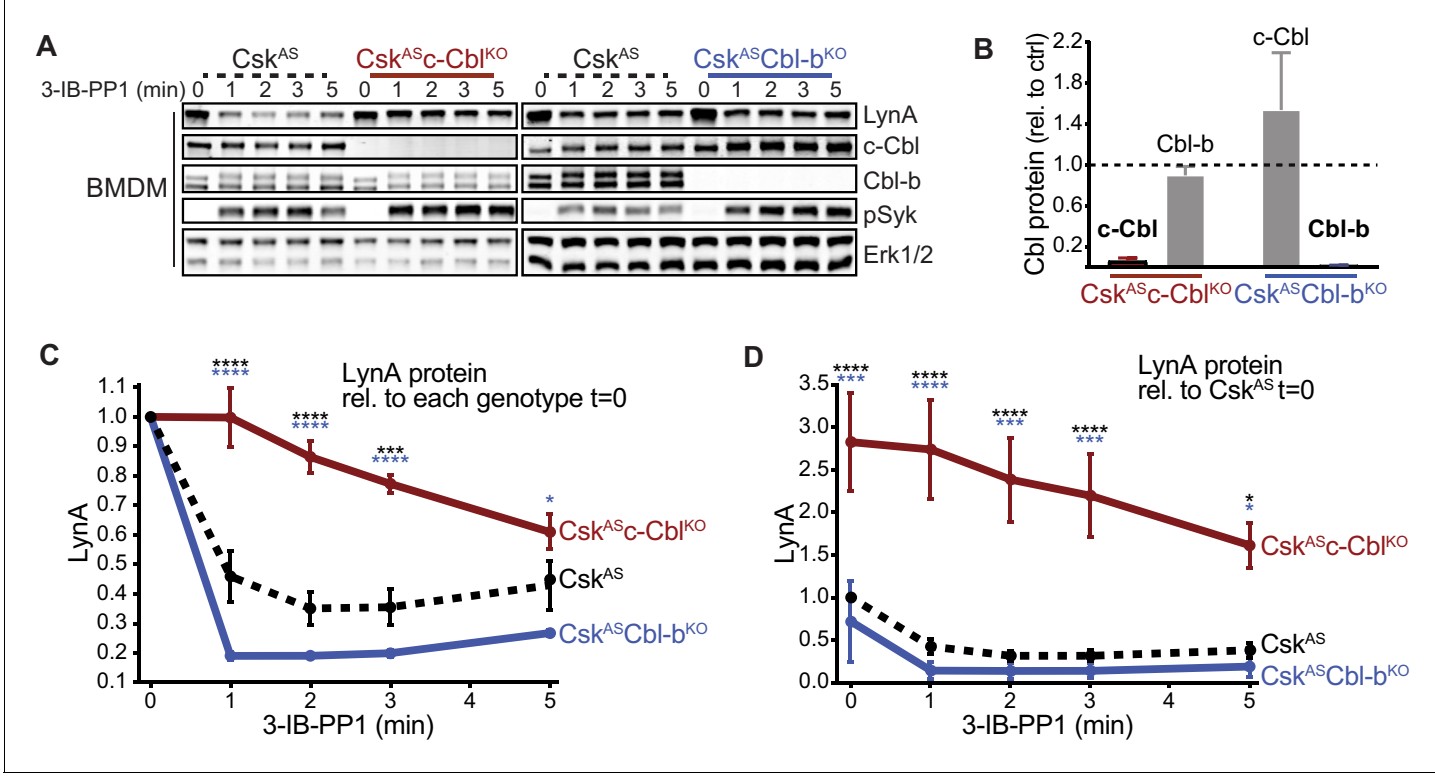

**Figure 1.** Loss of c-Cbl expression in macrophages leads to steady-state upregulation and delayed degradation of LynA protein. (**A**) Immunoblots showing LynA, c-Cbl, and Cbl-b protein in Csk[AS], Csk[AS]c-Cbl[KO], and Csk[AS]Cbl-b[KO] BMDMs treated with 3-IB-PP1 for the indicated times. Interdomain-B-phosphorylated Syk[pY352] (pSyk, *Yan et al., 2013*; *Wang et al., 2010*; *Law et al., 1996*), an SFK target and prerequisite for Syk activation, is shown as a control for 3-IB-PP1-initiated signaling; total Erk1/2 is shown as a loading control. (**B–D**) Densitometry quantification of relative levels of Cbl and LynA protein, corrected for total protein content using REVERT Total Protein Stain (TPS). (**B**) Quantification of c-Cbl (red) and Cbl-b (blue) in Csk[AS]c-Cbl[KO] and Csk[AS]Cbl-b[KO] BMDMs relative (rel.) to their steady-state levels in Csk[AS] BMDMs (dotted line). Expression of the other Cbl family member is shown in gray. Error bars reflect the standard error of the mean (SEM), n = 4 from three *Csk[AS]Cbl[-/-]* mice and n = 3 from two *Csk[AS]Cblb[-/-]* mice. (**C–D**) Quantification of LynA relative to the steady-state level in each genotype (**C**) or relative to the steady-state level in Csk[AS] BMDMs (**D**). SEM, n = 8 for Csk[AS], n = 5 from three *Csk[AS]Cbl[-/-]* mice, n = 3 from two *Csk[AS]Cblb[-/-]* mice. The significance (Sig.) from two-way ANOVA (ANOVA₂) with Tukey's multiple comparison test (-Tukey) are as follows: [Csk[AS]c-Cbl[KO] vs. Csk[AS] black asterisks], [Csk[AS]c-Cbl[KO] vs. Csk[AS]Cbl-b[KO], blue asterisks]; ****p<0.0001, ***p=0.0001–0.0005, * P=0.0171–0.0459. No significant difference was detected between other pairs (ns). Note: some of the error bars are smaller than the line width. Refer to *Figure 1—source data 1*. Paradoxical changes in Erk1/2 phosphorylation in Cbl-deficient cells are shown in *Figure 1—figure supplement 1*. Supporting siRNA studies are shown *Figure 1—figure supplement 2*.

DOI: https://doi.org/10.7554/eLife.46043.002

The following source data and figure supplements are available for figure 1:

**Source data 1.** Quantification of Cbl and LynA proteins in BMDMs lacking c-Cbl or Cbl-b.

DOI: https://doi.org/10.7554/eLife.46043.006

**Figure supplement 1.** Paradoxical changes in Erk1/2 phosphorylation in Cbl-deficient cells obscure the contribution of LynA to signaling.

DOI: https://doi.org/10.7554/eLife.46043.003

**Figure supplement 2.** siRNA knockdown of c-Cbl expression in macrophages leads to steady-state upregulation and delayed degradation of LynA protein.

DOI: https://doi.org/10.7554/eLife.46043.004

**Figure supplement 2—source data 1.** Quantification of Cbl and LynA proteins in BMDMs deficient in c-Cbl or Cbl-b.

DOI: https://doi.org/10.7554/eLife.46043.005

corresponding increase in c-Cbl expression as a consequence of Cbl-b knockdown (*Figure 1—figure supplement 2B*).

As in the knockout models, loss of c-Cbl but not Cbl-b resulted in impaired degradation of LynA in the first few minutes of 3-IB-PP1 treatment and increased steady-state expression of LynA protein. Without c-Cbl upregulation in Cbl-b-knockdown BMDMs, there was no increase in the LynA degradation efficiency or decrease in the steady-state level of LynA protein (*Figure 1—figure supplement*

2C–D), supporting our hypothesis that this observation in Csk[AS]Cbl-b[KO] BMDMs was c-Cbl-dependent. The effects of c-Cbl knockdown are blunted relative to Csk[AS]c-Cbl[KO] BMDMs, likely due to incomplete knockdown. As in the knockout experiments, LynA degradation in c-Cbl[KO] BMDMs was not fully eliminated but occurred on a much slower timescale, similar to the degradation of the other SFKs (*Freedman et al., 2015*).

Overall, we conclude that c-Cbl is solely responsible for the rapid and specific degradation of LynA, whereas other E3 ubiquitin ligases can complement the canonical SFK-binding and -polyubiquitinating function of c-Cbl (*Sanjay et al., 2001*; *Sanjay et al., 2006*), which mediates slower degradation of LynA and the other SFKs (*Freedman et al., 2015*).

### A tyrosine residue in the unique-region insert of LynA is required for its rapid degradation

Susceptibility to rapid, activation-induced degradation differentiates LynA from the splice variant LynB and the other abundant SFKs in macrophages, Hck (59 and 56 kDa transcripts) and Fgr (*Scapini et al., 2009*), which are degraded >10 fold more slowly than LynA during 3-IB-PP1 treatment (*Freedman et al., 2015*). The SFK FynT (Fyn) has also been reported to play an important role in macrophage inflammatory signaling (*Mkaddem et al., 2017*). We now confirm that, like Hck, Fgr, and LynB, macrophage Fyn is long-lived during 3-IB-PP1 treatment (*Figure 2A*), although we have been unable to detect Fyn activation in response to 3-IB-PP1 (*Freedman et al., 2015*). We also tested degradation of transfected LynA in comparison with endogenous Fyn and Lck in Jurkat T cells cotransfected with c-Cbl and a membrane-localized variant of Csk[AS] (memCsk[AS]) (*Schoenborn et al., 2011*), which sensitizes Jurkat cells to 3-IB-PP1, enabling synchronized SFK activation. As in macrophages, LynA is rapidly degraded in Jurkat cells treated with 3-IB-PP1, in contrast to longer-lived Lck (*Schoenborn et al., 2011*) and Fyn (*Figure 2B*). This highlights the unique susceptibility of LynA to rapid, c-Cbl-mediated degradation and suggests that cell types, such as T cells, that do not express Lyn may lack an analogous, rapid off-switch for SFK signaling.

To determine which residues in the LynA protein mediate the unique interaction with c-Cbl, we undertook a mutational analysis. We were, however, unable to achieve expression of LynA variants in Csk[AS] or Csk[AS]Lyn[KO] BMDMs (*Freedman et al., 2015*) by transfection, Amaxa nucleofection, or lentiviral transduction, likely due to toxicity of overexpressed Lyn. As an alternative approach, we expressed LynA ectopically in Jurkat T cells by transient cotransfection of His$_6$V5-tagged variants of

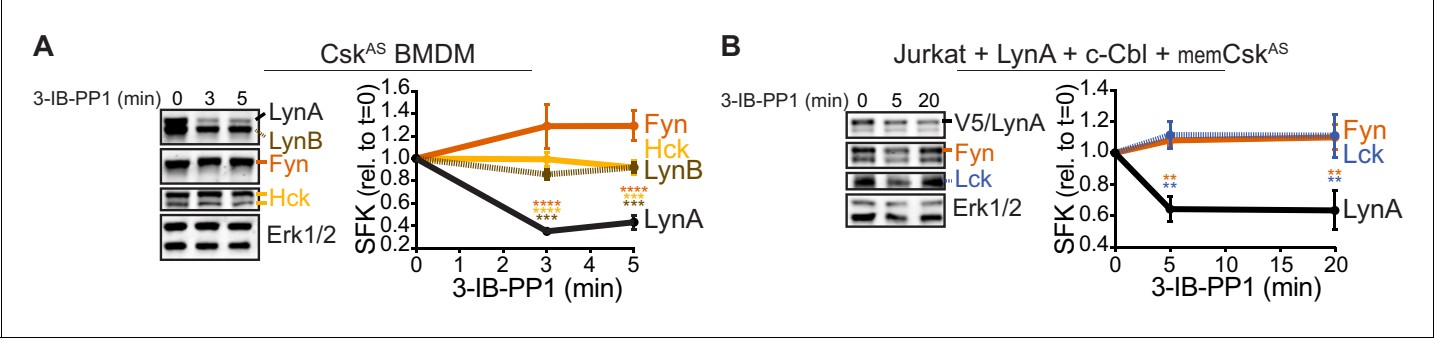

**Figure 2.** LynA is degraded more rapidly than Fyn and Lck during 3-IB-PP1 treatment. Immunoblots and quantification of SFK levels over the course of 3-IB-PP1 treatment. In both panels total Erk1/2 protein is shown as a loading control. Quantified values are corrected for total protein content (TPS) and reported relative to the steady-state level of each SFK. (**A**) Levels of Fyn (dark orange), Hck (light orange), LynA (black), and LynB (brown dashed) in 3-IB-PP1-treated Csk[AS] BMDMs. SEM, n = 3. Sig. from ANOVA$_2$-Tukey: [Fyn vs. LynA, dark orange asterisks], [Hck vs. LynA, light orange asterisks], [LynB vs. LynA, brown asterisks]; ****p<0.0001, ***p=0.0006–0.0009, **p=0.0034. (**B**) Levels of Fyn (dark orange) and Lck (blue dashed) and His$_6$V5-tagged LynA (black) in 3-IB-PP1-treated Jurkat T cells cotransfected with His$_6$V5-tagged LynA, memCsk[AS], and c-Cbl. SEM, n = 4. Sig. from ANOVA$_2$-Tukey: [Lck vs. LynA, blue asterisks], [Fyn vs. LynA, dark orange asterisks]; **p=0.0018–0.0048. Other pairs ns. Refer to *Figure 2—source data 1*.

DOI: https://doi.org/10.7554/eLife.46043.007

The following source data is available for figure 2:

**Source data 1.** Quantification of SFKs during 3-IB-PP1 treatment in mouse BMDMs and Jurkat cells.
DOI: https://doi.org/10.7554/eLife.46043.008

Lyn along with a Cbl-family ligase and memCsk$^{AS}$. As a proof of principle for this model, we first transfected Jurkat cells with wild-type LynA, memCsk$^{AS}$, and either empty vector (ctrl), c-Cbl, or Cbl-b plasmid DNA (*Figure 3A*), which increased c-Cbl expression 3-fold and Cbl-b expression 25-fold, respectively, over endogenous levels (*Figure 3B*). Overexpressing one Cbl family member did not affect the abundance of the other. Phosphorylation of Erk1/2, via the combined effects of transfected LynA and endogenous Lck, demonstrates that 3-IB-PP1 was applied to all transfectants (*Figure 3A*). As predicted, c-Cbl overexpression in Jurkat cells increased the rate of LynA degradation (30% depletion after 5 min treatment with 3-IB-PP1 in c-Cbl-overexpressing Jurkat cells versus no detectable depletion in either ctrl or Cbl-b-overexpressing cells) (*Figure 3C*). This observation is consistent with the knockout and siRNA experiments in BMDMs, demonstrating that Jurkat cells, which do not normally express Lyn, are capable of supporting rapid, c-Cbl-mediated degradation of LynA.

LynB, which lacks a 21 amino-acid insert found in the unique region of LynA (*Figure 3D*), was less efficiently degraded (10% depleted after 5 min treatment with 3-IB-PP1) than LynA (40% depleted after 5 min) in c-Cbl- and memCsk$^{AS}$-cotransfected Jurkat cells (*Figure 3E–F*). Again, these data mirror our findings in Csk$^{AS}$ BMDMs (*Freedman et al., 2015*) and confirm that the unique susceptibility of LynA to rapid degradation is preserved in the Jurkat model. We hypothesized that the c-Cbl recognition site in LynA must lie within the unique-region insert (residues 23–43), absent in LynB (*Yi et al., 1991*; *Stanley et al., 1991*), which contains a tyrosine residue reported to be phosphorylated in cancer cells (Y32; *Huang et al., 2013*; *Tornillo et al., 2018*), a predicted threonine phosphorylation site (T27; NetPhos 3.1, *Blom et al., 2004*) and a predicted lysine ubiquitination site (K40; UbPred, *Radivojac et al., 2010*) (*Figure 3D*).

Substituting tyrosine 32 with either alanine or phenylalanine blocked the rapid degradation of LynA (*Figure 3E*), reducing the depletion after 5 min treatment with 3-IB-PP1 from 40% (LynA) to 10% (LynA$^{Y32A}$) or 8% (LynA$^{Y32F}$), levels indistinguishable from LynB (10%) (*Figure 3E–F*). Substitution of Y32 did not alter the membrane localization of a unique-region construct at steady state or after 3-IB-PP1 treatment (*Figure 3—figure supplement 1*). Neither the predicted ubiquitination-site mutation (LynA$^{K40R}$) nor the predicted threonine phosphorylation site mutation (LynA$^{T27A}$) significantly affected the rate of LynA degradation (*Figure 3F*).

As in BMDMs, the efficiency of Lyn degradation correlated with its steady-state expression in Jurkat cells. LynB is 2-fold and LynA$^{Y32A}$ 3-fold more highly expressed than wild-type LynA, LynA$^{K40R}$, or LynA$^{T27A}$ at steady state, resulting in 3- to 4-fold more protein remaining after 20 min treatment with 3-IB-PP1 (*Figure 3—figure supplement 2*).

Overall, we conclude that the unique tyrosine residue Y32 in the LynA insert flags LynA for rapid, c-Cbl-mediated degradation and that this mechanism both determines the half-life of activated LynA protein during signaling and tunes its steady-state expression.

## Phosphorylation on tyrosine 32 targets activated LynA for polyubiquitination

Although phosphorylated LynA$^{Y32}$ has been found in liver and mammary tumor cells (*Huang et al., 2013*; *Tornillo et al., 2018*), it has been unclear whether this posttranslational modification is functionally relevant in hematopoietic cells and whether Y32 is phosphorylated as a consequence of SFK activation. To probe for Y32 phosphorylation, we performed quantitative analysis of LynA protein using targeted liquid-chromatography-coupled tandem mass spectrometry (LC-MS/MS) after immunoprecipitation of LynA from resting and 3-IB-PP1-treated BMDMs. We expected phosphorylation at this site to be transient due to the rapid degradation of LynA, and so we used a two-pronged approach to enrich potential Y32-phosphorylation: (i) to slow the rate of LynA polyubiquitination, we used BMDMs derived from *Csk$^{AS}$Cbl$^{+/-}$* mice, and (ii) we applied 3-IB-PP1 as a 15 s pulse treatment. At this early time point, all SFKs are expected to be activation-loop phosphorylated, with only 20% of LynA polyubiquitinated, as we have observed in wild-type BMDMs (*Freedman et al., 2015*).

Lysates from resting and 3-IB-PP1-treated BMDMs, normalized for total protein content, were subjected to immunoprecipitation using a LynA-specific antibody (*Freedman et al., 2015*) (*Figure 4A*, *Figure 4—figure supplement 1A*). Although some polyubiquitinated LynA species were generated, LynA was largely nonubiquitinated, as expected with low c-Cbl expression and an early time point of 3-IB-PP1 treatment. Bands from a Coomassie-stained gel containing nonubiquitinated and polyubiquitinated LynA were excised (*Figure 4B*, *Figure 4—figure supplement 1B*), normalized

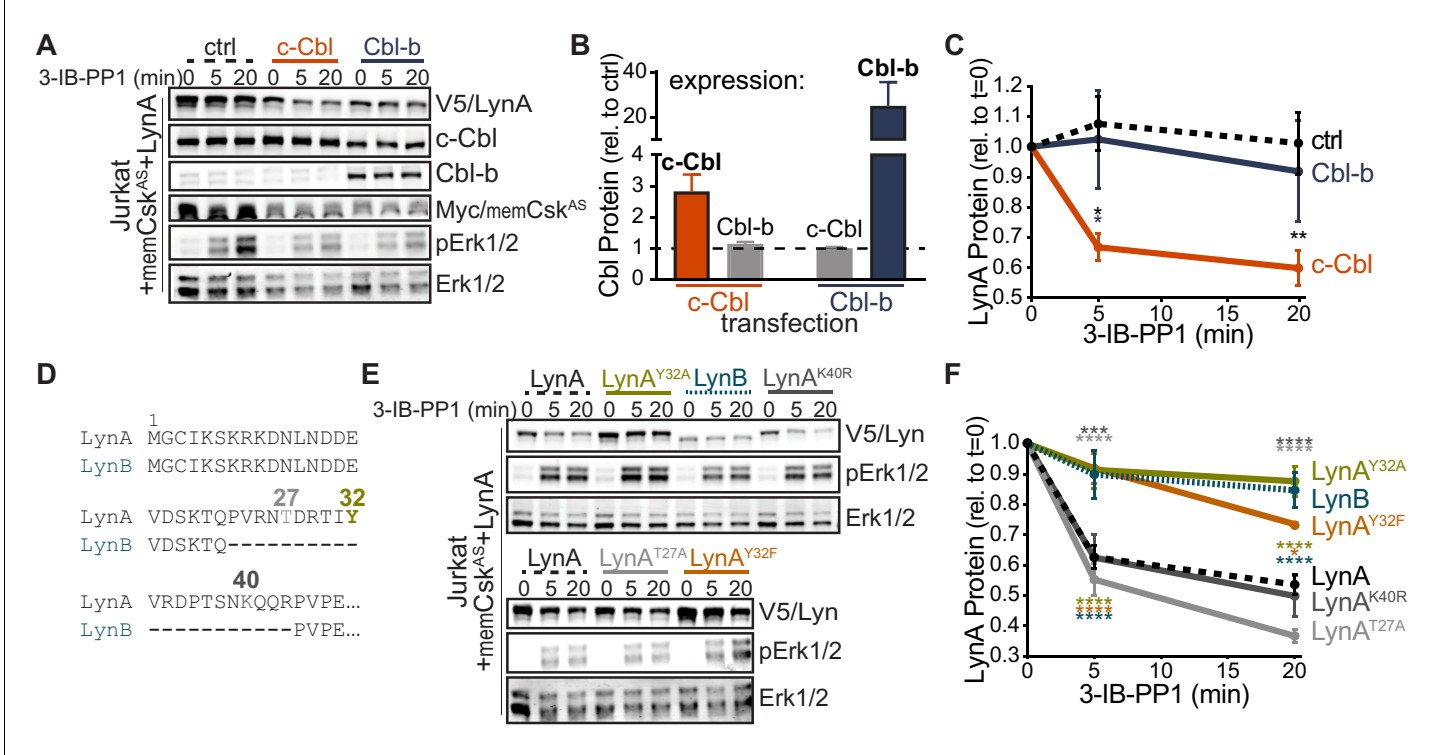

**Figure 3.** Unique-region tyrosine 32 is required for activation-induced degradation of LynA. (**A**) Immunoblots of Jurkat cells cotransfected with His$_6$V5-tagged LynA, Myc-tagged memCsk$^{AS}$, and either empty vector (ctrl), c-Cbl, or Cbl-b and treated with 3-IB-PP1. Phosphorylated Erk1/2$^{pT202/pY204}$ (pErk1/2) is shown as a qualitative control for 3-IB-PP1-initiated signaling from the combined effects of transfected LynA and endogenous Lck; total Erk1/2 is shown as a loading control. (**B–C**) Quantification of relative c-Cbl, Cbl-b, and His$_6$V5-tagged LynA protein, corrected for total protein content (TPS). SEM, n = 4. (**B**) Quantification of overexpressed c-Cbl (orange) and Cbl-b (dark blue) relative to endogenous (ctrl) levels (dotted line). Endogenous expression of the other Cbl family member in each condition is shown in gray. (**C**) Quantification of LynA protein during 3-IB-PP1 treatment relative to the steady-state level for transfections of empty-vector ctrl (black dotted), c-Cbl (orange), or Cbl-b (dark blue). Sig. from ANOVA$_2$-Tukey: [c-Cbl vs. ctrl, black asterisks], [c-Cbl vs. Cbl-b, blue asterisk]; **p=0.0092, *p=0.0270–0.0102. Other pairs ns. (**D**) N-terminal amino-acid sequences of mouse LynA and LynB (*Yi et al., 1991*; *Stanley et al., 1991*), including the 21 residue insert in the unique region of LynA, highlighting residues Y32 (olive), T27 (light gray), and K40 (dark gray). (**E**) Immunoblots of 3-IB-PP1-treated Jurkat cells cotransfected with c-Cbl, Myc-tagged memCsk$^{AS}$, and His$_6$V5-tagged Lyn constructs, including wild-type LynA, LynA$^{Y32A}$, LynB, LynA$^{K40R}$, LynA$^{T27A}$, and LynA$^{Y32F}$. (**F**) Quantification of protein levels of V5-tagged LynA (black dotted), LynA$^{Y32A}$ (olive), LynB (teal dotted), LynA$^{K40R}$ (dark gray), LynA$^{T27A}$ (light gray), and LynA$^{Y32F}$ (orange) over the course of 3-IB-PP1 treatment, corrected for total protein content (TPS) and reported relative to the steady-state level for each Lyn variant. SEM, n = 7 for LynA, n = 4 for LynA$^{Y32A}$ and LynB; n = 3 for LynA$^{K40R}$, LynA$^{T27A}$, and LynA$^{Y32F}$. Sig. from ANOVA$_2$-Tukey: [LynA$^{Y32A}$ vs. LynA, olive asterisks], [LynB vs. LynA, teal asterisks], [LynA$^{Y32F}$ vs. LynA, orange asterisks], [LynA$^{K40R}$ vs. LynA$^{Y32A}$, dark gray asterisks], [LynA$^{T27A}$ vs. LynA$^{Y32A}$, light gray asterisks], [LynB vs. LynA$^{K40R}$, p=0.013, p=0.01], [LynB vs. LynA$^{T27A}$, p<0001], [Lyn$^{Y32F}$ vs. LynA$^{K40R}$, p=0.012, 0.0157], [Lyn$^{T27A}$ vs. LynA$^{Y32F}$, p<0.0001], [Lyn$^{Y32A}$ vs. LynA$^{K10R}$, p<0.0001]; ****p<0.0001, ***p=0.0006, *p=0.0166. Other pairs ns. Note: the error bar for LynA$^{Y32F}$ is smaller than the line width. Refer to *Figure 3—source data 1*. Refer to *Figure 3—figure supplement 1* for microscopy showing localization of unique-region-only constructs of Lyn. Refer to *Figure 3—figure supplement 2* for quantification of relative expression levels of LynA variants and LynB.
DOI: https://doi.org/10.7554/eLife.46043.009

The following source data and figure supplements are available for figure 3:

**Source data 1.** Quantification of c-Cbl and Lyn proteins coexpressed in Jurkat cells.
DOI: https://doi.org/10.7554/eLife.46043.013
**Figure supplement 1.** The unique regions of LynA, LynB, and LynA$^{Y32A}$ are membrane-localized in resting and 3-IB-PP1-treated cells.
DOI: https://doi.org/10.7554/eLife.46043.010
**Figure supplement 2.** LynA$^{Y32A}$ and LynB are more highly expressed than wild-type LynA when expressed ectopically in Jurkat cells.
DOI: https://doi.org/10.7554/eLife.46043.011
**Figure supplement 2—source data 1.** Quantification of transfected Lyn constructs relative to wild-type LynA.
DOI: https://doi.org/10.7554/eLife.46043.012

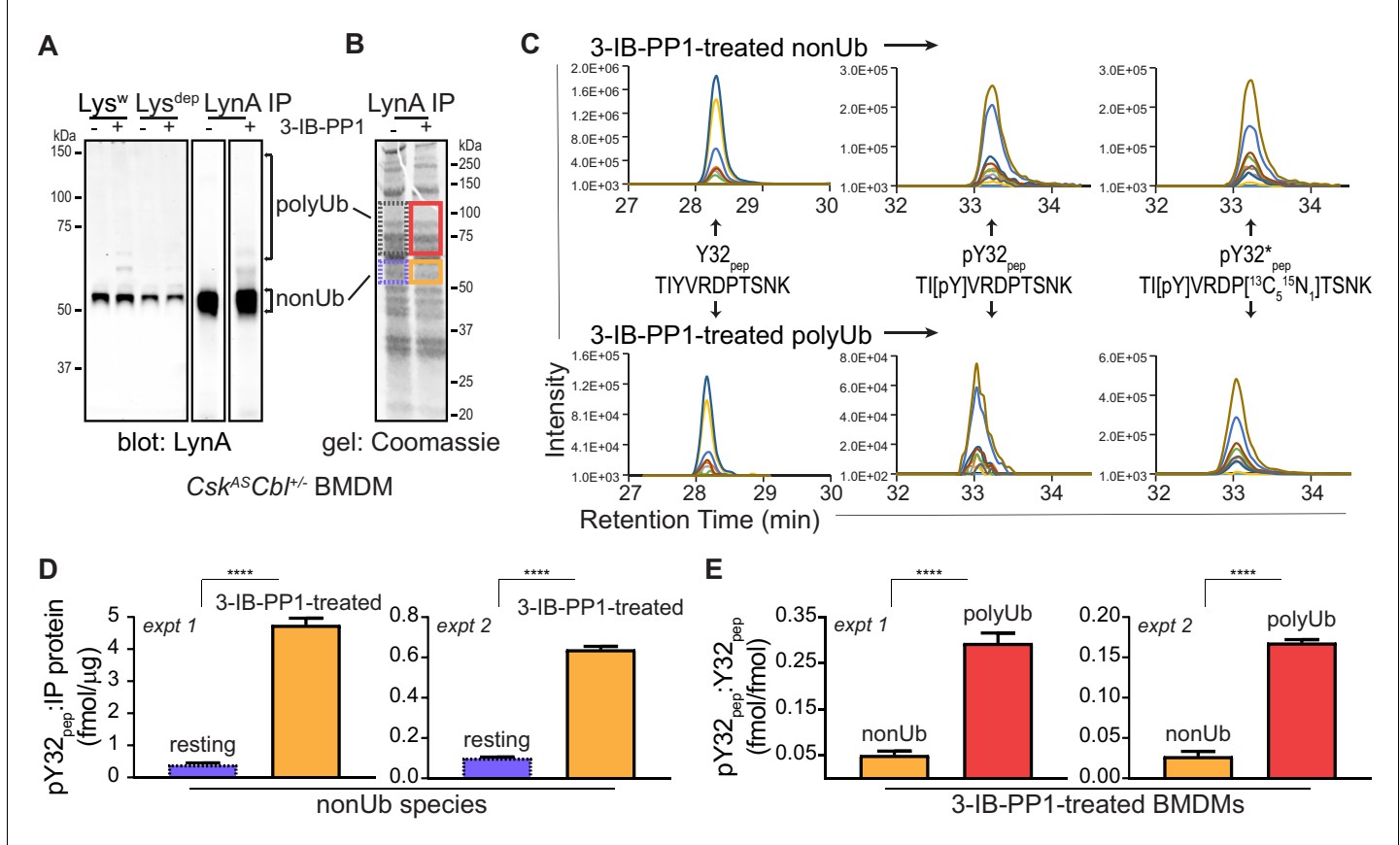

**Figure 4.** Tyrosine 32 is a site of activation-induced phosphorylation in macrophages. (**A**) Blots showing immunoprecipitation of LynA from BMDMs derived from $Csk^{AS}Cbl^{+/-}$ mice with (+) or without (-) a 15 s treatment with 3-IB-PP1. Nonubiquitinated (nonUb) and polyubiquitinated (polyUb) LynA species in whole-cell and immunodepleted lysate ($Lys^w$ and $Lys^{dep}$, respectively) and immunoprecipitate (IP) samples are shown. Uncropped immunoblots are shown in *Figure 4—figure supplement 1A*. (**B**) Coomassie-stained gel showing regions excised for LC-MS/MS analysis of polyUb LynA (higher boxes) and nonUb LynA (lower boxes); box colors correspond to the bar graphs below. Total IP protein content was quantified by applying densitometry to the whole lane and deriving a mass value via a BSA standard curve, shown in *Figure 4—figure supplement 1B*. Equivalent amounts of IP material were spiked with 125 fmol isotope-labeled control peptide ($pY32^*_{pep}$, a tryptic peptide containing phosphoY32) and subjected to in-gel trypsin digestion and LC-MS/MS. (**C**) Tryptic peptides $Y32_{pep}$ (IP-derived unphosphorylated peptide), $pY32_{pep}$ (IP-derived phosphopeptide), and added $pY32^*_{pep}$ were quantified by using parallel reaction monitoring (PRM) to collect MS/MS spectra for peptide parent ions 687.3267 and 690.3336 $m/z$, respectively. Extracted ion chromatograms (XICs) were derived from Skyline for MS/MS fragment ions corresponding to y and b fragments from Y32, pY32, and pY32* peptides and used to quantify the ratios of $pY32_{pep}:pY32^*_{pep}$ or $pY32_{pep}:Y32_{pep}$. Representative ion-annotated XICs are shown in *Figure 4—figure supplement 2*. Assignment of XIC peaks using synthetic peptides is shown in *Figure 4—figure supplement 3*. Detection of phosphopeptides in IP samples is shown in *Figure 4—figure supplement 4*. Peptide calibration curves are shown in *Figure 4—figure supplement 5*. (**D–E**) Quantitative analysis of Y32 phosphorylation in resting and 3-IB-PP1-treated BMDMs, including nonUb species from resting BMDMs (blue dotted), nonUb species from 3-IB-PP1-treated BMDMs (orange), and polyUb species from 3-IB-PP1-treated BMDMs (red). Although we analyzed a gel fragment that could contain polyUb species in resting BMDMs (gray dotted), this species was so rare as to be unquantifiable. Error bars represent the standard deviation (stdev) of n = 3 technical replicates. Data from two biological replicates are shown side by side. (**D**) Quantification of $pY32_{pep}$ in nonUb species from resting vs. 3-IB-PP1-treated BMDMs normalized to the total protein content of each IP. This analysis was used only for nonUb samples, which were roughly equimolar in resting and 3-IB-PP1-treated BMDMs. Sig. from two-tailed t tests for each experiment: [nonUb resting vs. nonUb 3-IB-PP1-treated, asterisks]; ****p<0.0001. (**E**) Quantification of $pY32_{pep}$ in nonUb vs. polyUb species in 3-IB-PP1-treated BMDMs normalized to the amount of unphosphorylated $Y32_{pep}$. Sig. from two-tailed t tests for each experiment: [nonUb resting vs. nonUb 3-IB-PP1-treated, asterisks]; ****p<0.0001. Refer to *Figure 4—source data 1*. Raw mass spectrometry files are available at the Panorama repository http://panoramaweb.org, http://panoramaweb.org/Freedman_LynA.url, ProteomeXchange ID: PXD014621).

DOI: https://doi.org/10.7554/eLife.46043.014

The following source data and figure supplements are available for figure 4:

**Source data 1.** Summary quantification of pY32 peptide from LC-MS/MS data.
DOI: https://doi.org/10.7554/eLife.46043.027

**Figure supplement 1.** Uncropped blots and BSA standard curve for mass spectrometry samples.

*Figure 4 continued on next page*

*Figure 4 continued*

DOI: https://doi.org/10.7554/eLife.46043.015

**Figure supplement 1—source data 1.** BSA standard curve for quantification of immunoprecipitated protein.
DOI: https://doi.org/10.7554/eLife.46043.016

**Figure supplement 2.** Annotated XICs.
DOI: https://doi.org/10.7554/eLife.46043.017

**Figure supplement 3.** PRM chromatograms and spectra for unphosphorylated and phosphorylated LynA Y32 tryptic peptides.
DOI: https://doi.org/10.7554/eLife.46043.018

**Figure supplement 4.** Y32 is the only site of phosphorylation detected in LynA Y32-containing tryptic peptide.
DOI: https://doi.org/10.7554/eLife.46043.019

**Figure supplement 5.** Peptide calibration curves.
DOI: https://doi.org/10.7554/eLife.46043.020

**Figure supplement 5—source data 1.** Standard curve for quantification of pY32 peptide relative to pY32* peptide in LynA immunoprecipitates.
DOI: https://doi.org/10.7554/eLife.46043.021

**Figure supplement 5—source data 2.** Standard curve for quantification of pY32 peptide relative to Y32 peptide in LynA immunopr.
DOI: https://doi.org/10.7554/eLife.46043.022

**Figure supplement 5—source data 3.** Quantification of pY32 peptide in nonUb LynA in resting BMDMs.
DOI: https://doi.org/10.7554/eLife.46043.023

**Figure supplement 5—source data 4.** Quantification of pY32 peptide in polyUb LynA in resting BMDMs.
DOI: https://doi.org/10.7554/eLife.46043.024

**Figure supplement 5—source data 5.** Quantification of pY32 peptide in nonUb LynA in 3-IB-PP1-treated BMDMs.
DOI: https://doi.org/10.7554/eLife.46043.025

**Figure supplement 5—source data 6.** Quantification of pY32 peptide in polyUb LynA in 3-IB-PP1-treated BMDMs.
DOI: https://doi.org/10.7554/eLife.46043.026

for protein content, and spiked with reference amounts of isotope-labeled control peptide corresponding to the tryptic fragment of Lyn encompassing pY32. These samples were then subjected to in-gel trypsin digestion and analyzed by LC-MS/MS.

Peptides corresponding to phosphorylated and unphosphorylated LynA Y32, including the isotope-labeled standard, were detected in nonubiquitinated and polyubiquitinated LynA species from 3-IB-PP1-treated cells (*Figure 4C*, *Figure 4—figure supplement 2*). We were able to resolve phosphorylated and unphosphorylated Y32 peptide and other potential phosphorylated species within the same tryptic peptide via their LC elution profiles: pY32 peptide coeluted with its isotopically labeled counterpart, and these were clearly separated from unphosphorylated and threonine-phosphorylated species (*Figure 4—figure supplement 3A*). The identity of each peptide was assigned via the MS/MS fragmentation pattern at the expected *m/z* (*Figure 4—figure supplement 3B–F*).

In the LynA immunoprecipitates the tryptic peptide spanning residue 32 was phosphorylated at Y32 but not at T30 or T37 (*Figure 4—figure supplement 4*). Search results in the PEAKS software package did not yield any matches for single or double threonine phosphorylation in combination with pY32, enabling complete analysis of LynA Y32 phosphorylation using an isotope-labeled pY32 peptide as the sole reference. Molar quantities of phosphorylated and unphosphorylated LynA Y32 peptide were derived by integrating the appropriate extracted ion chromatogram (XIC) peaks and correcting these values using standard curves with known molar ratios of unlabeled and isotope-labeled pY32 peptides (*Figure 4—figure supplement 5A*). For some analyses, a second calibration curve was used to determine the relative quantities of unphosphorylated Y32 peptide (*Figure 4—figure supplement 5B*).

We detected LynA Y32 phosphorylation in 3-IB-PP1-treated BMDMs in two independent experiments. After immunoprecipitation and sample processing, pY32 was 6- to 12-fold enriched in nonubiquitinated LynA from 3-IB-PP1-treated BMDMs relative to resting BMDMs (*Figure 4D*). LynA Y32 is therefore a site of inducible phosphorylation following activation of SFKs in primary macrophages. In resting BMDMs the vast majority of LynA is inactive (*Freedman et al., 2015*) and therefore protected from bulk degradation by lack of Y32 phosphorylation. Phosphorylated Y32 peptide was, however, detectable at low levels in resting BMDMs. This suggests that the suppression of steady-state LynA expression by c-Cbl (*Figure 1*, *Figure 1—figure supplement 2*) is attributable to mass action, driven by degradation of small quantities of basally active LynA.

Although phosphorylation of LynA Y32 was detected in activated, nonubiquitinated LynA, it was 6.0- to 6.4-fold higher in polyubiquitinated LynA (*Figure 4E*). It is therefore likely that phosphorylation of LynA Y32 is an early consequence of SFK activation and a prerequisite for rapid polyubiquitination by c-Cbl.

## Activated LynA induces its own degradation in trans

In macrophages and other hematopoietic cells SFKs are the first kinases to be activated upon ITAM-coupled receptor engagement, phosphorylating intracellular ITAMs, ITAM-associated Syk or Zap70 (*Yan et al., 2013*; *Wang et al., 2010*), and integrin-associated FAK or Pyk2 (*Lowell, 2011*; *Han et al., 2013*). We have reported that Syk and FAK are phosphorylated on activating tyrosine residues in Csk[AS] BMDMs treated with 3-IB-PP1 (*Freedman et al., 2015*). To test whether these downstream kinases might participate in negative feedback leading to LynA degradation, we treated Csk[AS] macrophages with competitive inhibitors of Syk (*Yamamoto et al., 2003*), FAK/Pyk2 (*Buckbinder et al., 2007*), or the SFKs themselves (*Hanke et al., 1996*) in combination with 3-IB-PP1. As expected, activation-loop phosphorylation of Erk1/2 and the SFKs was blocked in samples cotreated with the pan-SFK inhibitor PP2, activation-loop phosphorylation of Erk1/2 but *not* the SFKs was blocked in samples cotreated with the Syk inhibitor BAY-61–3606, and phosphorylation of Paxillin phosphorylation was blocked in samples cotreated with the FAK/Pyk2 inhibitor PF-431396 (*Figure 5A*).

LynA degradation was unaffected by Syk or FAK/Pyk2 inhibition but abrogated by SFK inhibition (*Figure 5A–B*). Given these data, we conclude that neither Syk, FAK, or Pyk2 nor any of their downstream targets participate in a negative-feedback loop leading to LynA degradation. This suggests that the only kinases upstream of Syk, FAK, and Pyk2, the SFKs themselves, induce LynA degradation. This tight feedback circuit could explain the strikingly short half-life of activated LynA.

To investigate the role of LynA kinase activity in its degradation, we turned again to Jurkat-cell transfections, testing two functionally impaired variants of LynA: LynA[T410K], which has disrupted substrate recognition (*Yokoyama and Miller, 1999*), and LynA[Y397F], which lacks the key autophosphorylation site that stabilizes the kinase active state and interacts with c-Cbl (*Chow and Veillette, 1995*; *Sanjay et al., 2001*; *Brown and Cooper, 1996*). To minimize interference from T-cell endogenous SFKs, we used Lck-deficient JCaM1.6 cells; previous work has shown that endogenous Fyn does not independently activate ITAM signaling in this cell line (*Straus, 1992*; *Goldsmith and Weiss, 1987*). When transfected into JCaM1.6 cells along with memCsk[AS] and c-Cbl, only catalytically competent His$_6$V5-tagged LynA was subject to rapid, 3-IB-PP1-induced degradation (*Figure 6A*); we did not detect degradation of the substrate-binding-impaired variant (T410K) or the activation-loop-mutated variant (Y397F). Zap70 phosphorylation was only visible in JCaM1.6 cells transfected with wild-type LynA, demonstrating the functional impairment of LynA[T410K] and LynA[Y397F].

We then transfected the same variants of LynA into Lck-expressing Jurkat cells (*Figure 6B*). The presence of phosphorylated Zap70 in all transfections demonstrates the expected activation of endogenous Lck in response to 3-IB-PP1 (*Schoenborn et al., 2011*; *Straus, 1992*; *Burgess et al., 1991*; *Samelson et al., 1990*). Activated Lck, however, was unable to induce degradation of LynA in trans, with no detectable loss of either catalytically-impaired variant (*Figure 6C*). To ensure that the substrate-binding and conformational elements altered in the T410K and Y397F variants did not affect our results, we confirmed that the active-site mutant LynA[K275R] was similarly resistant to 3-IB-PP1-induced degradation (*Figure 6—figure supplement 1*).

Observing that neither the Jurkat-cell SFKs (Lck/Fyn) nor the myeloid-cell SFK targets (Syk and FAK/Pyk2) participate in LynA degradation, we turned our focus to the major SFKs in macrophages. We showed above that wild-type LynA is degraded in 3-IB-PP1-treated Jurkat cells cotransfected with c-Cbl and memCsk[AS], suggesting that LynA provides the sole initiating signal to trigger its own degradation, in (i) its ability to induce the phosphorylation of LynA Y32 and (ii) its ability to phosphorylate and activate c-Cbl. To determine whether this effect could be mediated in trans by LynA and the other macrophage-expressed SFKs, we performed experiments in which LynA[T410K] (not degraded on its own during 3-IB-PP1 treatment) was coexpressed with kinase-active SFKs in Lck-deficient JCaM1.6 cells. LynA[T410K] degradation would indicate that other SFKs could induce both LynA Y32 phosphorylation in trans and c-Cbl activation (*Figure 7—figure supplement 1*). The composition of these transfected samples was assessed by immunoblotting for His$_6$V5-tagged LynA[T410K]

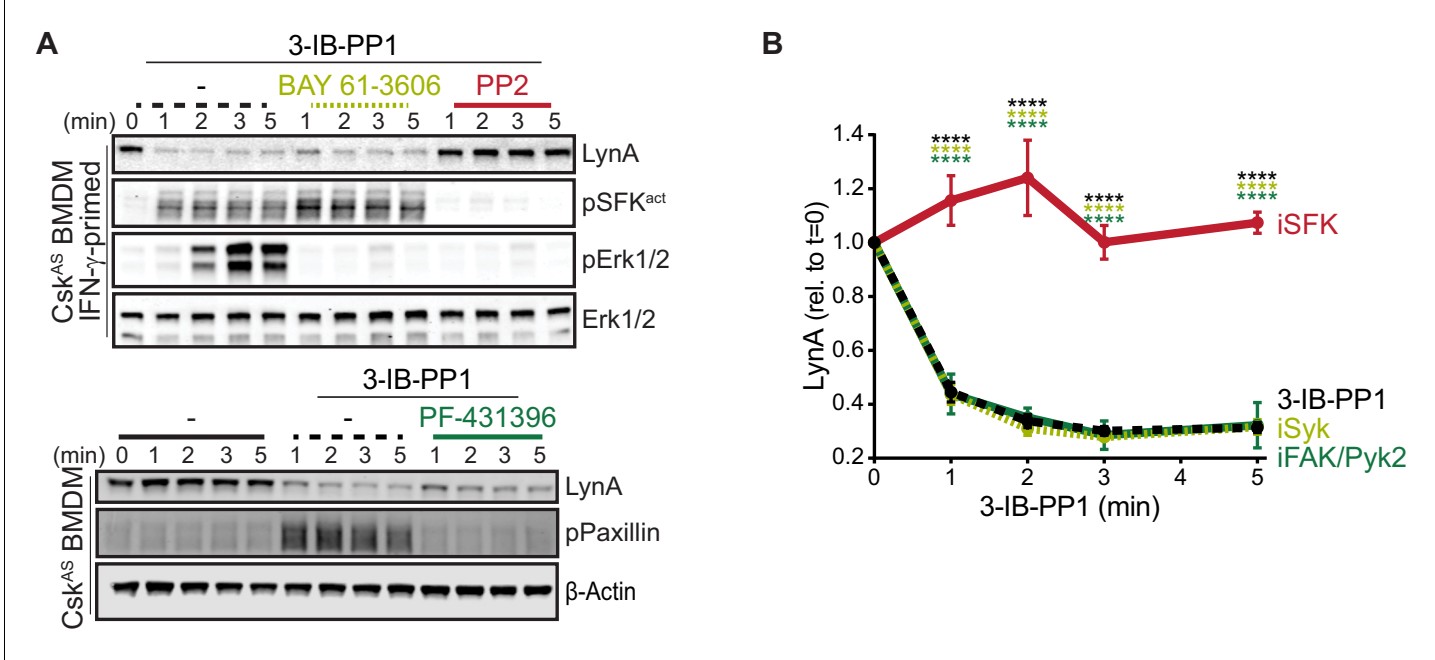

**Figure 5.** Syk, FAK, and Pyk2 kinase activities are not necessary for the rapid degradation of LynA. (**A**) Immunoblots showing expression of endogenous LynA protein in Csk[AS] BMDMs treated with 3-IB-PP1 only or cotreated with 3-IB-PP1 and the Syk inhibitor BAY 61–3606, the SFK inhibitor PP2, or the FAK/Pyk2 inhibitor PF-431396. Activated signaling proteins including SFKs (pSFK[act], activation-loop-phosphorylation corresponding to Src[pY416]), pErk1/2, and Paxillin[pY118] (pPaxillin) are shown as controls for inhibitor function; β-Actin and total Erk1/2 are shown as loading controls. (**B**) Quantification of LynA protein during 3-IB-PP1 treatment only (black dotted) or with the inhibitors BAY 61–3606 (iSyk, light olive), PP2 (iSFK, red), or PF-431396 (iFAK/Pyk2, green), corrected for total protein content (TPS) and reported relative to the steady-state level of LynA. SEM, n = 3. Sig. from ANOVA₂-Tukey: [3-IB-PP1 only vs. 3-IB-PP1+PP2, black asterisks], [3-IB-PP1+BAY 61–3606 vs. 3-IB-PP1+PP2, light olive asterisks], [3-IB-PP1+PF-431396 vs. 3-IB-PP1+PP2, green asterisks]; ****p<0.0001. Other pairs ns. Refer to *Figure 5—source data 1*.

DOI: https://doi.org/10.7554/eLife.46043.028

The following source data is available for figure 5:

**Source data 1.** Quantification of LynA degradation in BMDMs treated with 3-IB-PP1 and inhibitors.
DOI: https://doi.org/10.7554/eLife.46043.029

(for quantification in experimental samples) or wild-type His₆V5-tagged LynA (as a positive control). Wild-type LynA and LynA[Y32A] were cotransfected as untagged constructs, so they could be resolved independently of His₆V5-tagged LynA[T410K] on LynA immunoblots (*Figure 7A*). Cotransfected SFKs were visualized using antibodies specific for inactive and active SFKs, the V5 epitope tag, and/or individual Src family members (*Figure 7A*, *Figure 7—figure supplement 2A*). Cotransfected SFKs were classified by expression level, by ability to induce phosphorylation of the LynA[T410K] activation loop (pLynA[act]), phosphorylation of Zap70 (at an activating SFK substrate site in interdomain B, *Yan et al., 2013*; *Wang et al., 2010*), and phosphorylation of c-Cbl at Y731 (an SFK- but not Syk-dependent site of phosphorylation, *Buitrago et al., 2011*) (*Figure 7B*).

Cotransfected wild-type LynA, LynA[Y32A], and LynB were all capable of inducing the degradation of LynA[T410K] in trans. Where depletion of LynA[T410K] transfected alone was undetectable after 5 min of 3-IB-PP1 treatment, cotransfected wild-type LynA, LynA[Y32A], and LynB induced depletion of LynA[T410K] by 20%, 37%, and 40%, respectively; wild-type LynA alone was depleted by 40% at the same time point (*Figure 7C*). Although individual cells in a transiently transfected pool express varying levels of each construct, we were able to observe trends in the bulk expression levels of cotransfected SFKs. As reported by the LynA immunoblots for wild-type LynA and LynA[Y32A] (*Figure 7A*) and the V5 immunoblots for LynA and LynB (*Figure 7—figure supplement 2A*), LynA[Y32A] and LynB are more highly expressed than wild-type LynA. This is consistent with our previous observations (*Figure 3—figure supplement 2*). This difference in expression could explain the effectiveness of LynA[Y32A] and LynB in inducing degradation of LynA[T410K]; the unique-region substitution in LynA[Y32A]

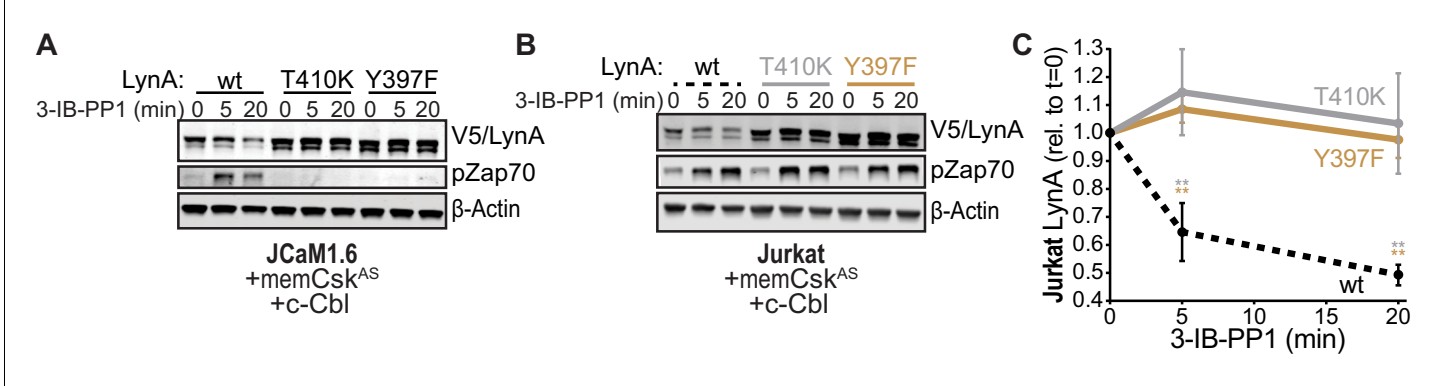

**Figure 6.** SFK activity is required for rapid degradation of LynA. (**A**) Immunoblots showing expression of transfected wild-type (wt), substrate-binding-impaired (T410K), or activation-loop-mutated (Y397F) His$_6$V5-tagged LynA protein coexpressed in JCaM1.6 cells with memCsk$^{AS}$ and c-Cbl and treated with 3-IB-PP1. Interdomain-B-phosphorylated Zap70$^{pY319}$ (pZap70), an SFK target and prerequisite for Zap70 activation, reflects the catalytic activity of LynA; β-Actin is shown as a loading control. (**B**) Overexpression of LynA variants in Jurkat cells. pZap70 reflects 3-IB-PP1-induced Lck activation. (**C**) Quantification of V5-tagged LynA variants wt (black dotted), T410K (gray), and Y397F (orange) in Jurkat cells during 3-IB-PP1 treatment, corrected for total protein content (TPS) and reported relative to the steady-state level for each variant of LynA. SEM, n = 3. Sig. from ANOVA$_2$-Tukey: [T410K vs. wt, gray asterisks], [Y397F vs. wt, orange asterisks]; \*\*p=0.0015–0.0083. Other pairs ns. Refer to *Figure 6—source data 1*. Refer to *Figure 6—figure supplement 1* for additional experiments with the catalytic-site mutant LynA$^{K275R}$.
DOI: https://doi.org/10.7554/eLife.46043.030

The following source data and figure supplements are available for figure 6:

**Source data 1.** Quantification of kinase-impaired LynA proteins expressed in Jurkat cells.
DOI: https://doi.org/10.7554/eLife.46043.033
**Figure supplement 1.** Catalytically dead LynA$^{K275R}$ is not targeted for rapid degradation.
DOI: https://doi.org/10.7554/eLife.46043.031
**Figure supplement 1—source data 1.** Quantification of LynA$^{K275R}$ protein in Jurkat cells during 3-IB-PP1 treatment.
DOI: https://doi.org/10.7554/eLife.46043.032

does not noticeably block this function or catalytic activity in general, at least compared to similarly expressed LynB. Potential differences in activity or signaling specificity, however, would be better investigated in a system that does not rely on overexpression.

Expression of Hck from a dual 59 kDa- and 56 kDa-expressing (Hck$^{59+56}$) plasmid induced depletion of LynA$^{T410K}$ by 17% after a 5 min treatment with 3-IB-PP1 (*Figure 7C*). Based on estimates from pSFK blots, both isoforms of Hck were expressed at high levels and pLynA$^{act}$, pZap70, and p-c-Cbl were strongly induced (*Figure 7A–B*). This activity, however, did not translate into a higher rate of LynA$^{T410K}$ degradation than the poorly expressed, cotransfected wild-type LynA (*Figure 7A–B*), suggesting that the ability to induce LynA degradation in trans depends on substrate specificity or other protein-protein interactions. Hck$^{59}$ alone was able to phosphorylate the activation loop of LynA$^{T410K}$ but unable to phosphorylate Zap70 or c-Cbl or induce degradation of LynA$^{T410K}$ (*Figure 7A,B,D*). This suggests that in the dual expression condition only the shorter isoform, Hck$^{56}$, was responsible for mediating phosphorylation of Zap70 and c-Cbl and degradation of LynA.

Endogenous Fyn in Jurkat cells is not activated during 3-IB-PP1 treatment (*Figure 7A*), and we were unable to increase the expression or 3-IB-PP1-induced activation of Fyn by cotransfection. We did, however, detect overexpression and 3-IB-PP1-induced activation of a cotransfected brain isoform of Fyn (FynB). Like Hck$^{59}$, FynB induced activation-loop phosphorylation of LynA$^{T410K}$ (*Figure 7B*) but did not phosphorylate Zap70 or c-Cbl or induce LynA$^{T410K}$ degradation (*Figure 7—figure supplement 2A*).

Although neither was able to induce degradation of LynA$^{T410K}$, Fgr and Lck were markedly different from Hck$^{59}$ and FynB in that, in addition to activation-loop-phosphorylating LynA$^{T410K}$, they were also capable of inducing Zap70 and c-Cbl phosphorylation (*Figure 7B,D*, *Figure 7—figure supplement 2A–B*). Again, this suggests that the failure of Lck and Fgr to mediate LynA degradation is due to substrate specificity rather than localization or lack of participation in ITAM signaling pathways.

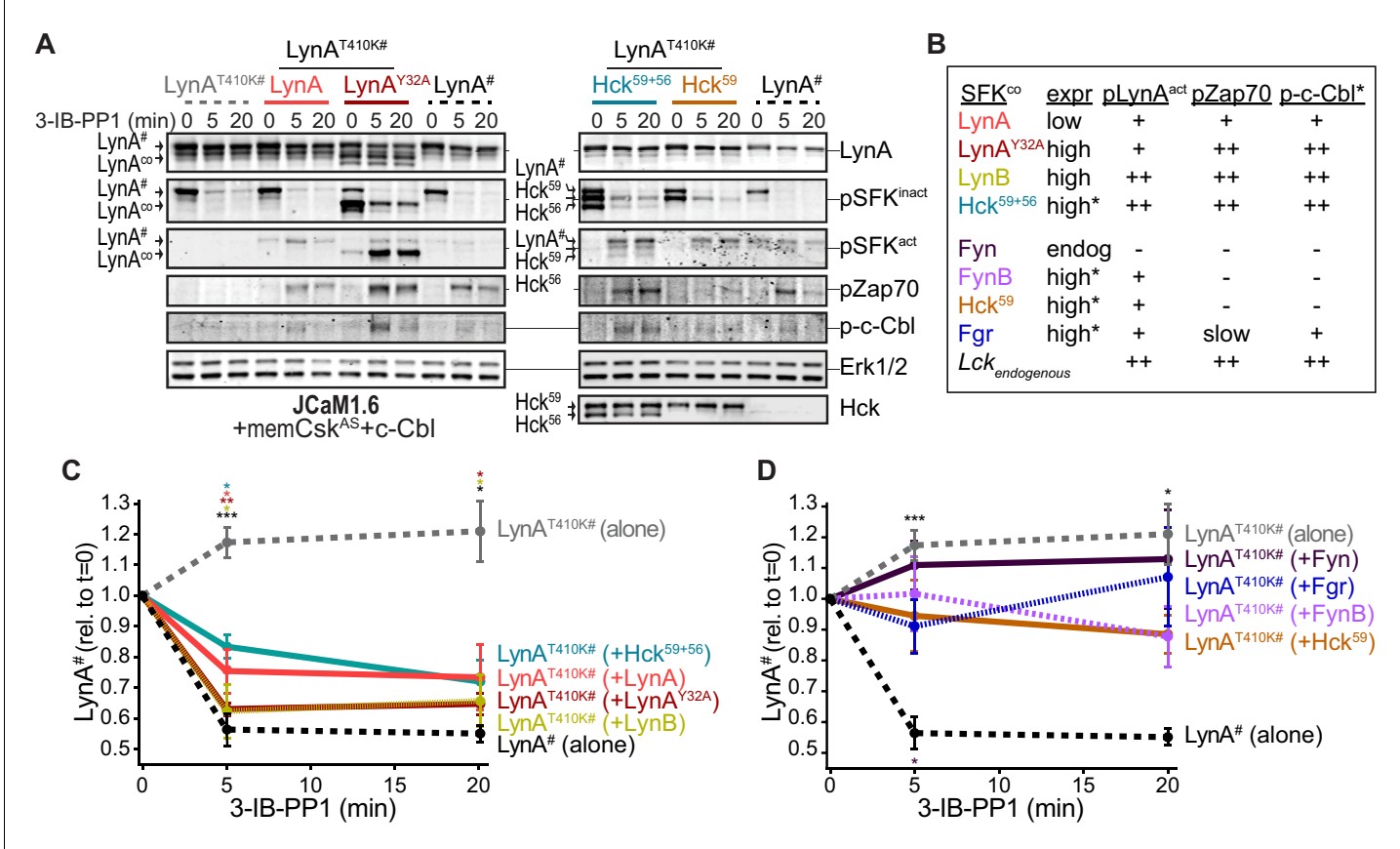

**Figure 7.** Cotransfected LynA, LynB, and Hck[56] can induce rapid degradation of LynA[T410K] in trans. (**A**) Representative immunoblots showing levels of His$_6$V5-tagged LynA[T410K] (#) cotransfected with empty vector or active SFK (co) in memCsk[AS]- and c-Cbl-expressing JCaM1.6 cells. Refer to *Figure 7—figure supplement 1* for an illustration of the experimental layout. Cotransfected SFKs include untagged LynA, untagged LynA[Y32A], His$_6$V5-tagged LynB, the long and short isoforms of Hck (Hck[59+56]), the long isoform of Hck (Hck[59]), FynT (Fyn), FynB, and Fgr. FynT is also expressed endogenously. His$_6$V5-tagged LynA[T410K#] (or wild-type LynA[#] in the positive control condition) was quantified from a total LynA blot (top panel). Colors correspond to later figure panels. Coexpressed SFKs are visualized via total SFK and pSFK[inact] (inhibitory-tail phosphorylated SFK, corresponding to Src[pY530]) blots as indicated. pSFK[act] (activation-loop phosphorylated, corresponding to Src[pY416]), pZap70, and p-c-Cbl blots are used to show 3-IB-PP1-induced activity of the cotransfected SFK; Erk1/2 is shown as a loading control. Refer to *Figure 7—figure supplement 2A* for immunoblots of LynB, Fgr, and Fyn cotransfections. When cotransfected with LynA[T410K], untagged wild-type LynA was expressed at very low levels, but its presence is inferred based on induction of Zap70 and c-Cbl phosphorylation with 3-IB-PP1 treatment. (**B**) Comparison of the expression (expr) levels of cotransfected SFKs (SFK[co]) and their ability to induce phosphorylation of the LynA[T410K] activation loop (pLynA[act], the highest band in the pSFK[act] blots), Zap70[pY319] and c-Cbl[pY731] (p-c-Cbl) following 3-IB-PP1 treatment. If expression could not be directly compared with the wild-type LynA in the positive-control sample, pSFK[inact] and/or pSFK[act] blots werecompared, with the caveat that the antibodies may not bind different SFKs equally well (marked with asterisks). Despite transfection, Fyn was expressed at endogenous levels (endog). Refer to *Figure 7—figure supplement 2B* for blots showing the effects of activating endogenous Lck. (**C–D**) Quantification of His$_6$V5-tagged wild-type LynA alone (black dotted) or His$_6$V5-tagged LynA[T410K] alone (gray dotted) or cotransfected with the kinase-active SFKs. All LynA quantifications (#) are corrected for total protein content (TPS) and reported relative to the steady-state level for each transfection condition. SEM, n = 5 for LynA[T410K] and LynA alone, SEM, n = 4 for all others. Sig. from ANOVA$_2$-Tukey: [LynA[T410K] vs. LynA black asterisks], LynA[T410K] vs. LynA[T410K]+LynA, coral asterisks], [LynA[T410K] vs. LynA[T410K]+LynA[Y32A], dark orange asterisks], [LynA[T410K] vs. LynA[T410K]+LynB, gold asterisks], [LynA[T410K] vs. LynA[T410K]+Hck[59+56], teal asterisks], [Lyn vs. LynA[T410K]+Fyn, dark purple asterisk]; ***p=0.0007, **p=0.0017, *p=0.0180–0.0466. Other pairs ns. (**C**) Quantification of His$_6$V5-tagged LynA[T410K] cotransfected with SFKs that induce degradation of LynA[T410K]. These include: untagged wild-type LynA (coral), untagged LynA[Y32A] (dark orange), His$_6$V5-tagged LynB (gold dotted), and Hck[59+56] (teal). (**D**) Quantification of His$_6$V5-tagged LynA[T410K] cotransfected with SFKs that do not induce degradation of LynA[T410K]: Hck[59] (tan), Fgr (blue dotted), Fyn (purple), and FynB (light purple). Refer to *Figure 7—source data 1*.

DOI: https://doi.org/10.7554/eLife.46043.034

The following source data and figure supplements are available for figure 7:

**Source data 1.** Quantification of LynA[T410K] coexpressed in Jurkat cells with other SFKs.
DOI: https://doi.org/10.7554/eLife.46043.037

**Figure supplement 1.** Assessing degradation of LynA mediated by SFKs in trans.

*Figure 7 continued on next page*

*Figure 7 continued*

DOI: https://doi.org/10.7554/eLife.46043.035

**Figure supplement 2.** Additional immunoblots from cotransfection experiments.

DOI: https://doi.org/10.7554/eLife.46043.036

Overall, these results show that LynA degradation can be induced by SFK activity in trans. Activated LynA, LynB, and Hck[56] can all mediate LynA degradation to some extent, but Lyn appears to do this most efficiently, due either to faster activation kinetics or more efficient phosphorylation of LynA and c-Cbl.

## Differential expression of c-Cbl tunes LynA protein levels and signaling in macrophages and mast cells

Analysis of data from the Immunological Genome Project shows that mRNA expression of Lyn and c-Cbl are differentially regulated across different cell types. Mast cells, for instance, express very low levels of *Cbl* mRNA (*Figure 8*) (*Heng and Painter, 2008*; *Dwyer et al., 2016*), which translates into a low level of c-Cbl protein expression (*Gustin et al., 2006*). *Cblb* mRNA, in contrast, is abundant in mast cells.

As in macrophages, Lyn is a key regulator of mast-cell ITAM signaling (*Shelby et al., 2016*), but the effect of disparate c-Cbl expression on LynA in the two cell types remains unknown. We generated bone-marrow-derived mast cells from Csk[AS] mice and performed surface marker analysis as we had for BMDMs (*Freedman et al., 2015*). Csk[AS] mast cells were homogeneous in their expression of the mast-cell markers FcεR1α and c-Kit (CD117) (*Kalesnikoff and Galli, 2011*) (*Figure 9A*).

We then used immunoblotting to compare the expression of c-Cbl, Cbl-b, and LynA in Csk[AS] mast cells and BMDMs and found that mast cells express very little c-Cbl (20% of the level in rested macrophages) and relatively high levels of Cbl-b (80% of the level in rested macrophages) (*Figure 9B–C*). Strikingly, low c-Cbl expression was accompanied by a complete resistance of LynA to degradation in the first 5 min of 3-IB-PP1 treatment (*Figure 9D*). Mast cells also had increased steady-state expression of LynA (30% higher than in rested BMDMs) (*Figure 9E*), despite low reported mRNA expression (*Figure 8*). These combined effects resulted in 3-4x more LynA protein in mast cells than in rested BMDMs and 2x more than in primed BMDMs between 3 and 5 min of 3-IB-PP1 treatment. Mast cells also had a stronger signaling response to 3-IB-PP1. Although the kinetics of Erk phosphorylation in mast cells varied, induction of pErk1/2 was consistently higher in mast cells than in macrophages, including in primed BMDMs (*Figure 9F*). We therefore hypothesized that cell-specific expression of c-Cbl tunes the abundance of LynA protein at steady state and the half-life of that protein during activation. Since mast cells are less dependent on c-Cbl for positive signaling (*e.g.* as a scaffold in PI3K activation), long-lived LynA potentiates a strong response to 3-IB-PP1. Thus mast cells bypass the LynA signaling checkpoint by maintaining low levels of c-Cbl and therefore high levels of activated LynA.

The relationship between c-Cbl expression, LynA degradation, and sensitivity to SFK-mediated signaling suggests that the responsiveness of myeloid cells can be tuned by changes in c-Cbl expression at the mRNA and/or protein level. As a direct test of this effect, we transfected mast cells derived from Csk[AS] bone marrow with a small activating (sa)RNA duplex (*Portnoy et al., 2011*; *Voutila et al., 2017*; *Voutila et al., 2012*) designed to increase the expression of *Cbl* mRNA. Transfection of this saRNA into mast cells increased the expression of c-Cbl protein an average of 2-fold relative to mock-transfected cells (*Figure 9G*). Csk[AS] mast cells were then treated with 3-IB-PP1 for up to 15 min to activate SFKs and assess LynA degradation and signaling potential as reported by pErk1/2 (*Figure 9H*). Even this modest increase in mast-cell c-Cbl expression increased LynA degradation in response to 3-IB-PP1 and suppressed downstream phosphorylation of Erk1/2 (*Figure 9I–J*). 3-IB-PP1 treatment for 15 min led to a 30% reduction of LynA protein in c-Cbl-overexpressing mast cells; LynA degradation at this time point was undetectable in control mast cells (*Figure 9I*). Correspondingly, Erk1/2 phosphorylation was completely suppressed in mast cells treated with c-Cbl saRNA relative to control mast cells (*Figure 9J*). Together, these data are consistent with a model in which differential expression of c-Cbl in myeloid cells regulates the activation response of LynA,

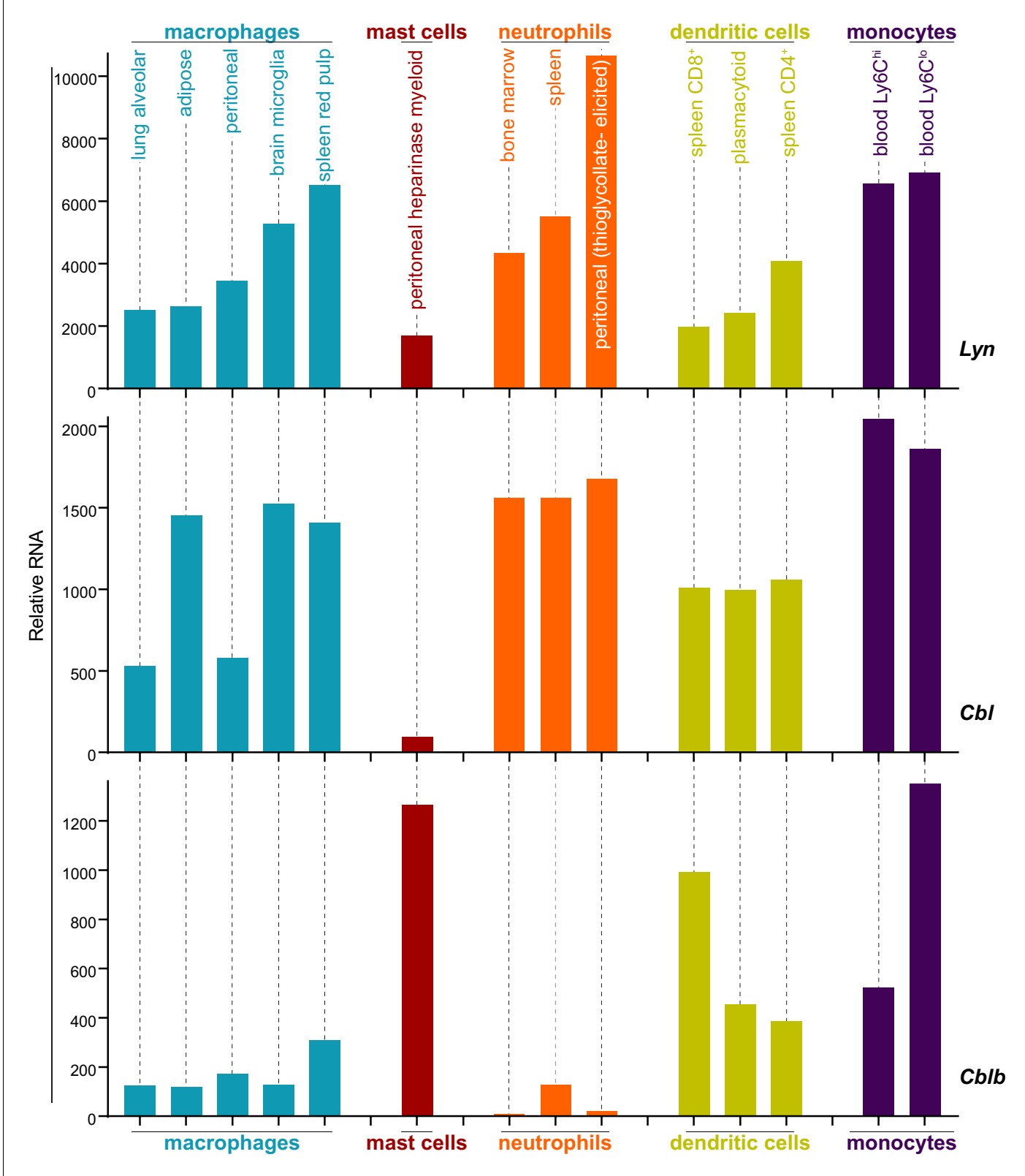

**Figure 8.** Expression levels of *Lyn*, *Cbl*, and *Cblb* are regulated differentially in myeloid cells. Mouse RNAseq data derived from the Immunological Genome Project (ImmGen, http://rstats.immgen.org/Skyline/skyline.html, ***Heng and Painter, 2008***; ***Dwyer et al., 2016***). *Lyn*, *Cbl*, and *Cblb* mRNA expression levels are shown for macrophages, mast cells, neutrophils, dendritic cells, and monocytes. Refer to ***Figure 8—source data 1***.
DOI: https://doi.org/10.7554/eLife.46043.038

*Figure 8 continued on next page*

*Figure 8 continued*

The following source data is available for figure 8:

**Source data 1.** Expression data from Immgen.

DOI: https://doi.org/10.7554/eLife.46043.039

working together with transcriptional regulation to tune myeloid-cell responsiveness to SFK-mediated signaling.

## Discussion

Polyubiquitination of signaling proteins by the Cbl family can lead to their degradation via the endo-lysosome or proteasome and to non-degradation-mediated inhibition (*Duan et al., 2004*). These E3 ligases must in turn be tightly regulated to prevent hyperresponsive or aberrant immune signaling while maintaining proper pathogen clearance (*Zhu et al., 2016*). Although highly homologous, c-Cbl and Cbl-b target different subsets of signaling proteins for inhibition and/or degradation (*Thien and Langdon, 2005*; *Mohapatra et al., 2013*), with c-Cbl polyubiquitinating the SFKs, including Lyn (*Kyo et al., 2003*; *Shao et al., 2004*), and Cbl-b acting on ITAM-coupled receptors, Syk, and other kinases (*Purev et al., 2009*; *Sohn et al., 2003*; *Gustin et al., 2006*; *Zhu et al., 2016*). c-Cbl binds via its phosphotyrosine-binding (PTB) domain to the SFK activation-loop phosphotyrosine (*Sanjay et al., 2001*) and via its PXXP motifs to the SFK SH3 domain (*Sanjay et al., 2006*). It is also an SFK substrate, requiring phosphorylation to become fully activated. These binding interfaces promote general c-Cbl interaction with all the SFKs. In this paper we report that c-Cbl targets LynA for rapid and specific degradation, a process that occurs with a half-life of a minute in BMDMs. In contrast, other E3 ligases contribute to the slower degradation mechanism shared among the other SFKs, including LynB, Fgr, Hck[59], Hck[56], Lck, FynB, and FynT, with the caveat that FynT does not seem to be activated in response to 3-IB-PP1.

c-Cbl is targeted to LynA via a noncanonical recognition site present in LynA but not the other SFKs. Although LynA and LynB have identical activation kinetics, lipidation sites, and activation-loop and SH3-domain sequences, LynA is degraded >10 fold more quickly than LynB in macrophages treated with the Csk[AS] inhibitor/SFK activator 3-IB-PP1 (20). Exploiting the similarity between LynA and LynB to make a limited set of point mutations, we have discovered that, while activation-loop phosphorylation and SH3-domain interactions mediate slower-phase targeting by c-Cbl and other E3 ligases, rapid degradation of LynA is triggered by an interaction peculiar to its unique-domain insert region.

We have discovered that phosphorylation of Y32 in the unique-region insert of LynA is required for its rapid degradation. This phosphorylation occurs within seconds of LynA activation in macrophages, and flags LynA for rapid, c-Cbl-mediated polyubiquitination. Phosphorylation of LynA Y32 had been observed in neuroblastoma cell lines upon receptor-tyrosine-kinase activation (*Palacios-Moreno et al., 2015*). In epidermoid carcinoma (A431) cells and breast tumor samples LynA Y32 is phosphorylated by the epidermal growth factor receptor (EGFR), allowing it to phosphorylate MCM-7 and stimulate cell proliferation (*Huang et al., 2013*). We now present evidence that LynA Y32 is phosphorylated in primary macrophages, where this potentially positive-regulatory function paradoxically induces LynA to trigger its own c-Cbl-mediated polyubiquitination and degradation, a process underlying its function as a rheostat controlling the LynA checkpoint.

Kinase-impaired variants of LynA are not rapidly degraded, a result confirmed in macrophages treated with the SFK inhibitor PP2. Catalytically active LynA and LynB, however, can induce degradation of catalytically impaired LynA in trans. Interestingly, the shorter isoform of Hck can also induce degradation of LynA, although less efficiently than Lyn itself. Degradation of LynA depends on a minimum of two phosphorylation events: the phosphorylation of LynA Y32 and activating phosphorylation of c-Cbl (*Feshchenko et al., 1998*; *Dou et al., 2012*). The ability of individual Src family members to induce LynA degradation necessarily relies on the efficiency of these two processes. In cotransfection experiments, we found that some SFKs (FynB, Hck[59], Fgr, and Lck) were unable to induce degradation of kinase dead LynA[T410K] despite an increase in their own activation-loop phosphorylation and their ability to phosphorylate the activation loop of LynA[T410K]. This suggests,

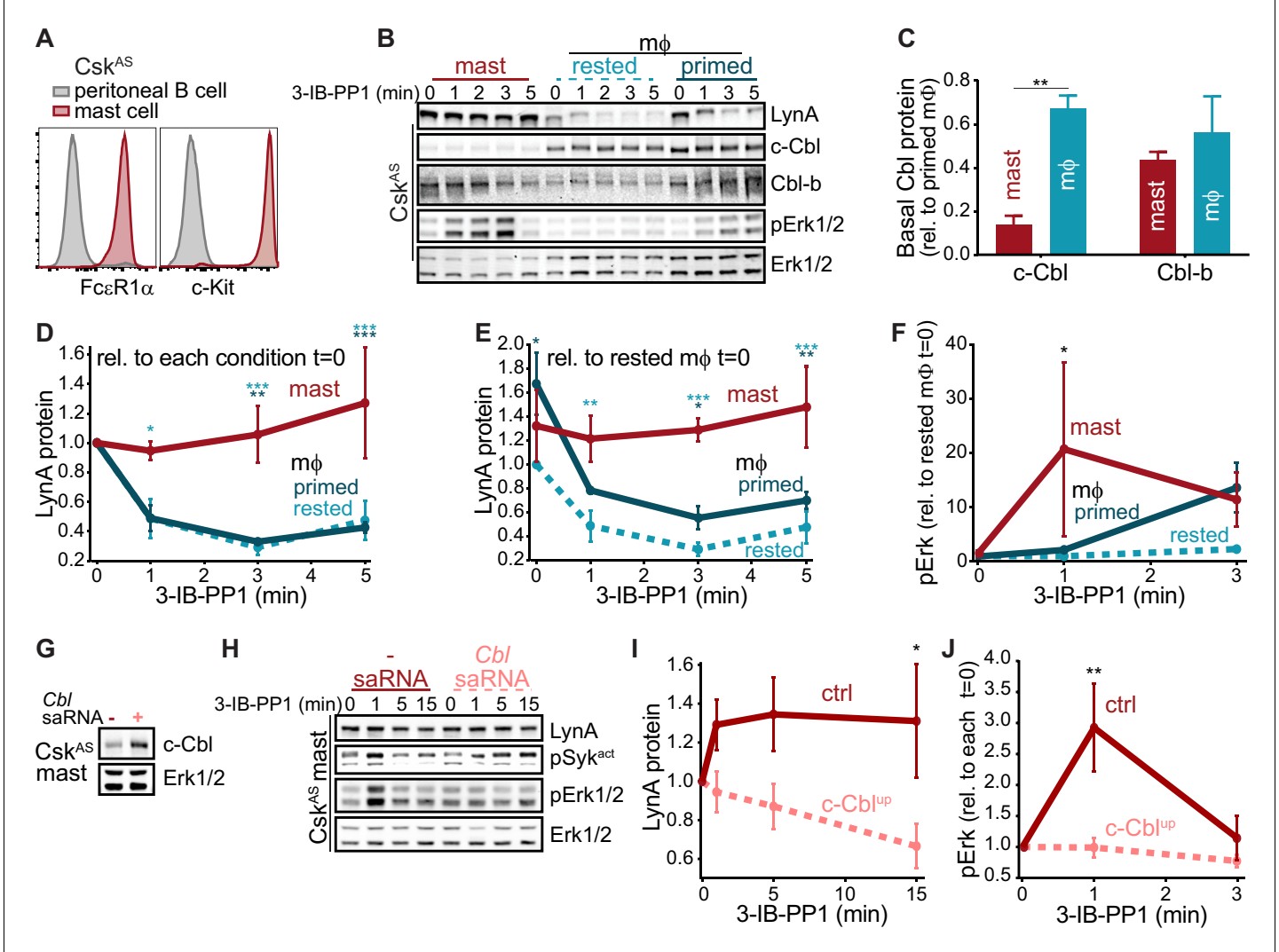

**Figure 9.** Differential expression of c-Cbl tunes LynA protein levels and SFK-mediated downstream signaling in macrophages and mast cells. (A) Flow cytometry showing surface expression of the mast-cell markers FcεR1α and c-Kit (CD117) in Csk^AS bone-marrow-derived mast cells (red) compared to peritoneal B cells (gray). (B) Immunoblots of Csk^AS mast cells and BMDMs (mφ) after resting or IFN-γ priming, showing the levels of LynA, c-Cbl, Cbl-b, and pErk1/2 during 3-IB-PP1 treatment; total Erk1/2 is shown as a loading control. (C–F) Quantification of c-Cbl, Cbl-b, LynA, and pErk in mast cells (red), IFN-γ-primed mφs (dark teal), and rested mφs (light teal). (C) Quantification of steady-state c-Cbl and Cbl-b protein in mast cells and rested mφs, corrected for total protein content (TPS) and reported relative to levels in primed mφs. SEM, n = 3. Sig. from one-way ANOVA with Sidak's multiple comparison test: [mast vs. rested mφs] **p=0.0035. Others pairs ns. (D–E) Quantification of LynA in mast cells and mφs during 3-IB-PP1 treatment, corrected for total protein content (TPS) and reported relative to the steady-state level of LynA in each cell type (D) or relative to the steady-state level of LynA in rested mφs (E). SEM, n = 3 for mast cells and primed mφs, n = 4 for rested mφs. Sig. from ANOVA₂-Tukey: [mast vs. rested mφs, light teal asterisks], [mast vs. primed mφs, dark teal asterisks]; ***p=0.0004–0.0008, **p=0.0025–0.0088, * P=0.0134–0.0466. Other pairs ns. (F) Quantification of pErk1/2 induction during 3-IB-PP1 treatment, corrected for total Erk1/2 expression and reported relative to the steady-state level of pErk1/2 in rested mφs. SEM, n = 3 for mast cells, n = 4 for primed mφs, n = 5 for rested mφs. Sig. from ANOVA₂-Sidak: [mast vs. rested mφs, asterisk]; *p=0.0330. Other pairs ns. (G) Immunoblots showing steady-state expression of c-Cbl in Csk^AS mast cells transfected in the presence or absence of Cbl-specific saRNA; Erk1/2 is shown as a loading control. (H) Immunoblots showing levels of LynA protein, pSyk, and pErk1/2 during 3-IB-PP1 treatment of Cbl-saRNA-transfected or mock-transfected mast cells. (I–J) Quantification of LynA and pErk in mast cells with (pink) and without (red) saRNA-induced upregulation of c-Cbl protein. SEM, n = 3. (I) Quantification of LynA protein during 3-IB-PP1 treatment, corrected for total protein content (TPS) and reported relative to the steady-state level for each transfection condition. Sig. from mixed-effects ANOVA analysis: *p=0.0229. (J) Quantification of pErk1/2 induction during 3-IB-PP1 treatment in saRNA-treated and mock-treated mast cells, corrected for total Erk1/2 expression and reported relative to the steady-state level for each condition. SEM, n = 3 for mast cells. Sig. from ANOVA₂-Sidak: **p=0.0045. Refer to **Figure 9—source data 1**.
DOI: https://doi.org/10.7554/eLife.46043.040
The following source data is available for figure 9:

*Figure 9 continued on next page*

*Figure 9 continued*

**Source data 1.** Comparison of mast cells and macrophages.
DOI: https://doi.org/10.7554/eLife.46043.041

perhaps surprisingly (*Lock et al., 1991*; *Carréno et al., 2000*), that these kinases are not localized away from LynA or otherwise lacking in activity. Instead, it is likely that the induction of LynA degradation occurs via specific protein-protein interactions and/or substrate recognition interfaces. Unlike FynB and Hck[59], Fgr and Lck were able to phosphorylate c-Cbl, despite an earlier report that c-Cbl is a poor substrate for Lck and Zap70 (*Feshchenko et al., 1998*). On the other hand, Syk and FynT have both been reported to phosphorylate and activate c-Cbl (*Feshchenko et al., 1998*), but LynA degradation in 3-IB-PP1-treated BMDMs is unaffected by Syk inhibition and Fyn is not activated during 3-IB-PP1 treatment. Again, we conclude that the substrate preferences of the Src family members are tightly regulated, contradicting the assumption that downstream players in the ITAM signaling pathway are activated en bloc by semi-promiscuous kinases.

Rapid degradation of LynA is easily observed upon bulk SFK activation by 3-IB-PP1 but also affects the steady-state level of LynA protein, likely due to the selective degradation of small amounts (*Freedman et al., 2015*) of basally active LynA over time. Although the unique region of LynA lacks a c-Cbl consensus recognition motif (*Thien and Langdon, 2005*), pY32 could serve as a docking site for the PTB domain of c-Cbl. In this ultrafast feedback process, activation-loop- and Y32-autophosphorylated LynA could efficiently phosphorylate and activate c-Cbl and trigger its own degradation, potentially then releasing c-Cbl for other positive- and negative-regulatory signaling functions. Earlier work has shown that although the interactome of transfected LynA[Y32F] in mammary adenocarcinoma (MDA-MB-231) cells more closely resembles that of wild-type LynA than LynB, some LynA interactions are lost (*Tornillo et al., 2018*). It is also possible that, by analogy to a similarly situated but non-homologous phosphorylation site in Hck (pY29) (*Johnson et al., 2000*), phosphorylation of LynA Y32 could be directly activating. The unique regions of LynA and LynB make distinct SH3-domain contacts (*Teixeira et al., 2018*), and phosphorylation could alter these interactions and increase kinase activity. Hyperactivated LynA could then efficiently phosphorylate and activate nearby or pre-complexed c-Cbl, triggering its own degradation more aggressively than do the other SFKs.

LynA and LynB have been reported to interact with different subsets of proteins in mast cells and in triple-negative breast cancer cells, with LynA signaling via ITAM (*Alvarez-Errico et al., 2010*), cytoskeletal, and proliferative (*Tornillo et al., 2018*) pathways and LynB initiating negative feedback via ITIMs and phosphatases (*Alvarez-Errico et al., 2010*). In mammary epithelial cells the ratio of LynA and LynB is actively regulated by Epithelial Splicing Regulatory Protein 1 (ESRP1). LynA-upregulated tumors have the more invasive phenotype (*Tornillo et al., 2018*). The positive-regulatory roles reported for LynA in mast and tumor cells complements our own observations that LynA degradation can block macrophage signaling through the Erk1/2, Akt, and NFAT pathways, which cannot be rescued by active LynB (*Freedman et al., 2015*).

Aberrant activation driven by ITAM signaling pathways in macrophages is a known driver of autoimmune and inflammatory disease, with activated macrophages accumulating in chronically inflamed tissues in the absence of an active infection (*Wynn et al., 2013*). Regulation of LynA protein expression and deactivation kinetics by c-Cbl could, via the LynA checkpoint, prevent the initiation of pathological signaling. The threshold for macrophage activation is modulated by changes to their local environment. For example, IFN-γ and LPS polarize macrophages for a pro-inflammatory response, whereas IL-4 and IL-13 polarize macrophages for tissue repair (i.e. collagen deposition) and inflammatory resolution (*Galli et al., 2011*; *Bosurgi et al., 2017*). We have reported that IFN-γ decreases the macrophage signaling threshold in part by increasing the expression of LynA (*Freedman et al., 2015*). Dynamic changes in SFK and c-Cbl levels could modulate the macrophage activation threshold, ensuring that macrophages respond appropriately during times of infection (low threshold) and limiting aberrant activation in response to cellular debris and small-scale antibody complexes during inflammatory resolution (high threshold).

Like macrophages, mast cells reside in nearly every tissue and perform environment-specific functions in addition to sensing non-self (*Galli et al., 2011*; *Gentek et al., 2018*). While macrophages

have distinct anti-inflammatory roles as professional phagocytes in the silent clearance of apoptotic cells and agents of wound healing, mast cells are constitutively primed for ITAM-induced triggering, releasing preformed granules that contain inflammatory cytokines, chemokines, prostaglandins, and proteases. Pursuant to these differing functions, macrophages continuously gauge ITAM ligand valency (*i.e.* particle size) and have a relatively high basal threshold for inflammatory activation (*Goodridge et al., 2011*; *Freedman et al., 2015*), while mast cells can be triggered by small-scale receptor clustering induced by low-valency or even monovalent FcεR-IgE-allergen complexes (*Andrews et al., 2009*; *Felce et al., 2018*). One striking difference between the macrophage and mast-cell ITAM regulatory machinery is that mast cells express almost no c-Cbl. In spite of low reported mRNA levels (ImmGen; *Heng and Painter, 2008*; *Dwyer et al., 2016*), mast cells constitutively express a high level of LynA protein, likely due to impaired steady-state degradation of basally active LynA. Furthermore, LynA is not degraded upon activation and 3-IB-PP1 triggers a stronger Erk phosphorylation response in the absence of ITAM-receptor ligation in mast cells than in rested macrophages.

In summary, we describe a model in which regulation of LynA and c-Cbl levels helps to determine an immune cell's potential for inflammatory ITAM signaling. Regulated degradation tunes down the LynA rheostat, blocking cell signaling via the LynA checkpoint at low cellular doses of LynA and overriding the LynA checkpoint at higher doses. In macrophages this occurs on a continuum where LynA dose is highest in cells rendered more reactive by IFN-γ priming (polarization). Mast cells express almost no c-Cbl and maintain high levels of LynA at steady state and over time, consistent with permissive signaling in the absence of a large-scale ITAM receptor clustering event nucleated by multivalent receptor ligation. Appropriate regulation of immune receptor thresholds is critical for maintaining the function of innate immune cells; threshold dysregulation can lead to chronic feedback loops that drive inflammatory signaling in autoimmune disease and conversely tumor-supporting immunosuppressive signaling (*Wynn et al., 2013*). Elucidating the mechanisms by which the LynA checkpoint is regulated may allow us to tune immune-cell sensitivity and reprogram pathological cells.

# Materials and methods

**Key resources table**

| Reagent type (species) or resource | Designation | Source or reference | Identifiers | Additional information |
|---|---|---|---|---|
| Genetic reagent (*M. musculus*) | Csk$^{AS}$ | PMID: 21917715 | MGI: 5563915 | Dr. Arthur Weiss (University of California San Francisco) |
| Genetic reagent (*M. musculus*) | Cbl$^{-/-}$ | PMID: 17868091 | MGI: 2180576 | Dr. Erik Peterson (University of Minnesota) |
| Genetic reagent (*M. musculus*) | Cblb$^{-/-}$ | PMID: 10646609 | MGI: 2180572 | Dr. Michael Farrar (University of Minnesota) |
| Cell line (*H. sapiens*) | Jurkat | American Type Culture Collection | Cat#: TIB-152, RRID: CVCL_0367 | Dr. Arthur Weiss (University of California San Francisco) |
| Cell line (*H. sapiens*) | JCaM1.6 | American Type Culture Collection | Cat#: CRL-2063, RRID: CVCL_0354 | Dr. Yoji Shimizu (University of Minnesota) |
| Recombinant DNA reagent (plasmid) | (mouse) LynA-V5-His6 | This paper | | Progenitor from Dr. Juan Rivera (NIH) Vector: pEF6/V5-His |
| Recombinant DNA reagent (plasmid) | (mouse) LynB-V5-His6 | This paper | | Progenitor from Dr. Juan Rivera (NIH) Vector: pEF6/V5-His |

*Continued on next page*

*Continued*

| Reagent type (species) or resource | Designation | Source or reference | Identifiers | Additional information |
|---|---|---|---|---|
| Recombinant DNA reagent (plasmid) | (mouse) LynA$^{Y32A}$-V5-His6 | This paper | | Progenitor from Dr. Juan Rivera (NIH) Vector: pEF6/V5-His |
| Recombinant DNA reagent (plasmid) | (mouse) LynA$^{K40R}$-V5-His6 | This paper | | Progenitor from Dr. Juan Rivera (NIH) Vector: pEF6/V5-His |
| Recombinant DNA reagent (plasmid) | (mouse) LynA$^{T27A}$-V5-His6 | This paper | | Progenitor from Dr. Juan Rivera (NIH) Vector: pEF6/V5-His |
| Recombinant DNA reagent (plasmid) | (mouse) LynA$^{Y32F}$-V5-His6 | This paper | | Progenitor from Dr. Juan Rivera (NIH) Vector: pEF6/V5-His |
| Recombinant DNA reagent (plasmid) | (mouse) LynA$^{T410K}$-V5-His6 | This paper | | Progenitor from Dr. Juan Rivera (NIH) Vector: pEF6/V5-His |
| Recombinant DNA reagent (plasmid) | (mouse) LynA$^{Y397F}$-V5-His6 | This paper | | Progenitor from Dr. Juan Rivera (NIH) Vector: pEF6/V5-His |
| Recombinant DNA reagent (plasmid) | (mouse) LynA$^{K275R}$-V5-His6 | This paper | | Progenitor from Dr. Juan Rivera (NIH) Vector: pEF6/V5-His |
| Recombinant DNA reagent (plasmid) | (mouse) LynA | This paper | | Progenitors from Dr. Juan Rivera (NIH) Vector: pEF6/V5-His |
| Recombinant DNA reagent (plasmid) | (mouse) LynA$^{Y32A}$ | This paper | | Progenitors from Dr. Juan Rivera (NIH) Vector: pEF6/V5-His |
| Recombinant DNA reagent (plasmid) | His6-Xpress-c-Cbl (human) | PMID: 17110436 | | Dr. Arthur Weiss (University of California San Francisco) Vector: pEF |
| Recombinant DNA reagent (plasmid) | Xpress-Cbl-b (human) | PMID: 17110436 | | Dr. Arthur Weiss (University of California San Francisco) Vector: pEF |
| Recombinant DNA reagent (plasmid) | (mouse) Hck$^{p59kDa}$ | This paper | | Dr. Clifford Lowell (University of California San Francisco) Vector: pMIG-W |
| Recombinant DNA reagent (plasmid) | (mouse) Hck$^{p59+p56kDa}$ | This paper | | Dr. Clifford Lowell (University of California San Francisco) Vector: pMIG-W |
| Recombinant DNA reagent (plasmid) | (mouse) FynT | This paper | | Dr. Clifford Lowell (University of California San Francisco) Vector: pUC18 |
| Recombinant DNA reagent (plasmid) | (human) FynB | This paper | | Dr. Clifford Lowell (University of California San Francisco) Vector: pLNCX |
| Recombinant DNA reagent (plasmid) | (mouse) Fgr | This paper | | Vector: pEF |
| Recombinant DNA reagent (plasmid) | UR(LynA)-eGFP | This paper | | vector: pcDNA3.1 |
| Recombinant DNA reagent (plasmid) | UR(LynB)-eGFP | This paper | | vector: pcDNA3.1 |

*Continued on next page*

*Continued*

| Reagent type (species) or resource | Designation | Source or reference | Identifiers | Additional information |
|---|---|---|---|---|
| Recombinant DNA reagent (plasmid) | UR(LynA$^{Y32A}$)-eGFP | This paper | | vector: pcDNA3.1 |
| Recombinant DNA reagent (plasmid) | eGFP | This paper | | Dr. Michael Farrar (University of Minnesota) vector: pMIGR |
| Sequence based reagent | *Cbl* siRNA | This paper | 5'-CCUACCAGGACAU UCAGAAAGCUUU-3' 5'-AAAGCUUUCUGAA UGUCCUGGUAGG-3' | |
| Sequence based reagent | *Cblb* siRNA | This paper | 5'-CUGACUUCUUG GUAUCUGAUAUATA-3' 5'-UAUAUAUCAGAUAC CAAGAAGUCAGGU-3' | |
| Sequence based reagent | *Cbl* saRNA | This paper | 5'-UCAAUUCUAG AUAAAGGCG-3' 5'-CGCCUUUAUCU AGAAUUGA-3' | |
| Antibody | Mouse anti-β-Actin (8H10D10) | Cell Signaling | Cat. #: 3700 RRID: AB_2242334 | WB (1:2000) |
| Antibody | Rabbit anti-c-Cbl (D4E10) | Cell Signaling | Cat. #: 8447 RRID: AB_10860763 | WB (1:2000) |
| Antibody | Rabbit anti-Cbl-b (D3C12) | Cell Signaling | Cat. #: 9498 RRID: AB_2797707 | WB (1:2000) |
| Antibody | Rabbit anti-p-c-Cbl (phosphoTyr$^{731}$) | Cell Signaling | Cat. #: 3554 RRID: AB_2070452 | WB (1:1000) |
| Antibody | Rabbit anti-pErk1/2 (phosphoThr$^{202}$/ phosphoTyr$^{204}$, D13.14.4E) | Cell Signaling | Cat. #: 4370 RRID: AB_10694057 | WB (1:2000) |
| Antibody | Mouse anti-Erk1/2 (3A7) | Cell Signaling | Cat. #: 9107 RRID: AB_10695739 | WB (1:2000) |
| Antibody | Mouse anti-Fgr (6G2) | ProMab | Cat. #: 20318 RRID: AB_2802161 | WB (1:2000) |
| Antibody | Rabbit anti-Fyn | Cell Signaling | Cat. #: 4023 RRID:AB_10698604 | WB (1:2000) |
| Antibody | Goat anti-Hck (M-28) | Santa Cruz Biotechnology | Cat. #: sc-1428 RRID: AB_2114872 | WB (1:1000) |
| Antibody | Rabbit anti-Lck (D88) XP(R) | Cell Signaling | Cat. #: 2984 RRID: AB_2136313 | WB (1:2000) |
| Antibody | Rabbit anti-LynA (C13F9) | Cell Signaling | Cat. #: 2796 RRID: AB_2138391 | WB (1:2000) |
| Antibody | Mouse anti-LynA + LynB (Lyn-01) | abcam | Cat. #: ab1890 RRID: AB_1204641 | WB (1:1000) |
| Antibody | Mouse anti-Myc (9B11) | Cell Signaling | Cat. #: 2276 RRID: AB_2148465 | WB (1:2000) |
| Antibody | Rabbit anti-pPaxillin (phosphoTyr$^{118}$) | Cell Signaling | Cat. #: 2541 RRID: AB_2174466 | WB (1:2000) |
| Antibody | Rabbit anti-pSFK$^{act}$ (phosphoTyr$^{416}$) | Cell Signaling | Cat. #: 2101 RRID: AB_331697 | WB (1:2000) |
| Antibody | Rabbit anti-pSFK$^{inact}$ (phosphoTyr$^{530}$) | Life Technologies/ThermoFisher | Cat. #: 44–912 RRID: AB_2533793 | WB (1:1000) |

*Continued on next page*

*Continued*

| Reagent type (species) or resource | Designation | Source or reference | Identifiers | Additional information |
|---|---|---|---|---|
| Antibody | Rabbit anti-pZap70/pSyk (phospho Tyr$^{319(Zap)/352(Syk)}$, 65E4) | Cell Signaling | Cat. #: 2717 RRID: AB_2218658 | WB (1:2000) |
| Antibody | Rabbit V5-Tag (D3H8Q) | Cell Signaling | Cat. #: 13202 RRID: AB_2687461 | WB (1:2000) |
| Antibody | Mouse anti-V5 | Invitrogen | Cat. #: R960-25 RRID: AB_2556564 | IP (1:200) |
| Antibody | Donkey anti-rabbit IgG 800CW | LI-COR | Cat. #: 925–32213 RRID: AB_2715510 | WB (1:20000) |
| Antibody | Donkey anti-rabbit IgG 680LT | LI-COR | Cat. #: 925–68023 RRID: AB_10706167 | WB (1:20000) |
| Antibody | Donkey anti-mouse IgG 800CW | LI-COR | Cat. #: 925–32212 RRID: AB_2716622 | WB (1:20000) |
| Antibody | Donkey anti-mouse IgG 680LT | LI-COR | Cat. #: 925–68022 RRID: AB_10715072 | WB (1:20000) |
| Antibody | Donkey anti-goat IgG 800CW | LI-COR | Cat. #: 926–32214 RRID: AB_621846 | WB (1:20000) |
| Antibody | Anti FcεRIa (FITC) | BioLegend | Cat. #: 134305 RRID: AB_1626102 | Flow cytometry (1:100) |
| Antibody | Anti c-Kit (APC) | BioLegend | Cat. #: 105811 RRID: AB_313220 | Flow cytometry (1:100) |
| Peptide, recombinant protein | recombinant mouse IFN-g | PeproTech | Cat. #:AF-315–05 | 25 U/mL |
| Peptide, recombinant protein | recombinant mouse IL-3 | PeproTech | Cat. #: 213–13 | 10 ng/mL |
| Purified protein | Trypsin, sequencing grade | Promega | Cat. #: V5111 | |
| Commercial assay | Pierce BCA Protein Assay | ThermoFisher | Cat. #: 23225 | |
| Chemical compound, drug | 3-IB-PP1 | Kevan Shokat (University of California, San Francisco) | | 10 µM |
| Chemical compound, drug | PP2 | ThermoFisher | Cat. #: PHZ1223 | 20 µM |
| Chemical compound, drug | Bay 61–3606 | EMD Millipore | Cat. #: 574714 | 1 µM |
| Chemical compound, drug | PF-431396 | Tocris | Cat. #: 4278/10 | 10 µM |
| Software | Prism | GraphPad | RRID: SCR_002798 | |
| Software | Image Studio | LI-COR | RRID:SCR_013715 | |
| Software | Illustrator | Adobe | RRID:SCR_010279 | |
| Software | Photoshop | Adobe | RRID:SCR_014199 | |
| Software | Fiji | | RRID:SCR_002285 | |

## Mice

C57BL/6-derived *Csk*$^{AS}$ mice are hemizygous for the *Csk*$^{AS}$ BAC transgene on a *Csk*$^{-/-}$ background, as described previously (*Freedman et al., 2015*; *Tan et al., 2014*). *Csk*$^{AS}$*Cblb*$^{-/-}$(*Csk*$^{-/-}$) mice were generated by crossing *Cblb*$^{-/-}$ female mice from M. Farrar (University of Minnesota) (*Chiang et al., 2000*) with *Csk*$^{+/-}$ and *Csk*$^{AS}$ male mice from our colony and then crossing *Csk*$^{+/-}$*Cblb*$^{-/-}$ with *Csk*$^{AS}$*Csk*$^{+/-}$*Cblb*$^{-/-}$ mice. Due to sterility of *Cbl*$^{-/-}$ male mice, we were not able to produce a

sustained lineage of $Csk^{AS}Cbl^{-/-}(Csk^{-/-})$ mice, but we obtained three individuals by crossing $Cbl^{-/-}$ female mice from E. Peterson (University of Minnesota) with $Csk^{+/-}$ and $Csk^{AS}$ male mice from our colony and then crossing $Csk^{+/-}Cbl^{+/-}$mice with $Csk^{AS}Cbl^{+/-}$ mice. All mice were housed in specific pathogen-free conditions and genotyped using real-time PCR (Transnetyx, Inc, Memphis, TN). All animal use complies with University of Minnesota (UMN) and National Institutes of Health (NIH) policy (Animal Welfare Assurance Number A3456-01). UMN is accredited by AAALAC, experiments involving mice were approved by the UMN Institutional Animal Care and Use Committee (IACUC, protocol # 1603-33559A). Animals are kept under supervision of a licensed doctor of veterinary medicine and supporting veterinary staff under strict NIH guidelines.

## Jurkat cell lines and transfection

The Jurkat T-cell strains Clone E6-1 (*Weiss and Stobo, 1984*) and JCaM1.6 (Lck-deficient) (*Straus, 1992*; *Goldsmith and Weiss, 1987*) were gifts from the laboratories of A. Weiss (University of California, San Francisco) and Y. Shimizu (University of Minnesota), respectively. Both cell lines were authenticated by STR profiling and tested negative for mycoplasma (ATCC, Manassas, Virginia). Jurkat cell lines were cultured in RPMI-1640 medium supplemented with 5–10% fetal calf serum (Omega Scientific, Inc, Tarzana, CA) and 2 mM glutamine, penicillin and streptomycin (Sigma-Aldrich, St. Louis, MO) as described previously (*Phee et al., 2005*). Jurkat and JCaM1.6 cells were transiently transfected via electroporation, as described previously (*Phee et al., 2005*). Briefly, cells were grown overnight in antibiotic-free RPMI-1640 medium supplemented with 10% fetal bovine serum (Omega Scientific) and 2 mM glutamine (RPMI10). Batches of 15 M cells were resuspended in RPMI10 with 10–15 µg plasmid DNA per construct. Cells were rested, electroporated at 285 V for 10 ms in a BTX square-wave electroporator (Harvard Apparatus, Holliston, MA), resuspended in RPMI10, and allowed to recover overnight. One million live cells were then resuspended in phosphate-buffered saline (PBS), rested for 30 min at 37°C, and stimulated.

## Microscopy

Jurkat cells were transfected with memCsk$^{AS}$, c-Cbl, and LynB (to ensure a response to 3-IB-PP1) along with constructs encoding the unique regions of LynA, LynB and LynA$^{Y32A}$ fused to eGFP. A construct containing eGFP alone was used as a control. 24 hr after transfection, cells were filtered to remove clumps of dead cells and resuspended in PBS containing Hoechst Stain (Thermo Fisher) and incubated for 30 min at 37°C in a 96-well plate. Cells were then stimulated with 3-IB-PP1 for 5 min before quenching with ice-cold PBS and collected by centrifuging for 5 min x 1500 rpm at 4°C. Cells were resuspended in ice-cold PBS and coverslipped on glass slides directly before imaging. Imaging was performed on a Leica DM600B epifluorescence microscope using DAPI and GFP channels. Exposure times for GFP were set to normalize the brightness of the GFP across the different transfection conditions, as cell-to-cell expression was variable. Images were pseudocolored and merged using ImageJ software (NIH) with the Fiji plugin (*Schindelin et al., 2012*).

## DNA constructs and mutagenesis

Plasmids containing c-terminally His$_6$V5-tagged mouse LynA and LynB were gifts from J. Rivera/R. Suzuki (National Institutes of Health) (*Alvarez-Errico et al., 2010*). Myc-tagged mouse memCsk$^{AS}$ (also referred to as Lck11-CskAS, *Schoenborn et al., 2011*) and Xpress-tagged human c-Cbl and Cbl-b (*Myers et al., 2006*) were gifts from A. Weiss (University of California, San Francisco). Untagged mouse Hck$^{56+59}$, Hck$^{56}$, Fgr, and FynT (Fyn), and human FynB were gifts from C. Lowell (University of California, San Francisco.) Site-directed mutagenesis (QuikChange Lightning, Agilent Technologies, Santa Clara, CA) was used to prepare the point mutants referenced in the main text and to introduce stop codons removing epitope tags from Fgr, LynA, and LynA$^{Y32A}$. Sequences of all constructs were verified by Sanger sequencing (GENEWIZ, South Plainfield, NJ).

## Preparation and flow cytometry of myeloid cells

BMDMs were prepared using standard methods (*Zhu et al., 2008*). Briefly, bone marrow was extracted from femura and tibiae of mice aged 6–8 weeks. After hypotonic lysis of erythrocytes, BMDMs were derived on untreated plastic plates (BD Falcon, Sigma-Aldrich) by culturing in Dulbecco's Modified Eagle Medium (DMEM, Corning Cellgro, Corning, NY) containing approximately 10%

heat-inactivated fetal calf serum (Omega Scientific) (DMEM10) with 0.11 mg/ml sodium pyruvate (Corning), 2 mM penicillin/streptomycin/L-glutamine (Sigma-Aldrich), and 10% CMG-14–12-cell-conditioned medium as a source of M-CSF (*Takeshita et al., 2000*). After 6 or 7 days cells were resuspended in enzyme-free ethylenediaminetetraacetic acid (EDTA) buffer and replated in untreated 6-well plates (BD Falcon, Sigma-Aldrich) at 1 M cells per well in unconditioned medium with or without priming in 25 U IFN-γ (PeproTech, Rocky Hill, NJ). Bone-marrow-derived mast cells were prepared as described previously (*Kalesnikoff and Galli, 2011*), similarly to BMDMs except for the substitution of 10 ng/ml IL-3 (PeproTech) for CMG-14–12-conditoned medium and a culture time of at least 5 weeks. Mast cells were subjected to flow-cytometry analysis (LSRFortessa, Becton Dickinson, Franklin Lakes, NJ) and found to be uniformly positive for FcεRIα (Mar-1, FITC-labeled, #134305) and c-Kit (2B8, APC-labeled, #105811), both from BioLegend (San Diego, CA). All myeloid cells were rested or primed overnight before stimulation.

## siRNA knockdown of *Cbl* and *Cblb*

The siRNA sequence for knockdown of mouse *Cbl* was adapted from a human siRNA construct (A. Weiss, University of California, San Francisco). *Cblb* siRNA was designed using the Integrated DNA Technologies double-stranded siRNA design tools (IDT, Skokie, IL). Control double-stranded RNA from IDT was not predicted to be complementary to any sequence in the human or mouse transcriptome. The effective siRNA sequences from among those we tested are: *Cbl* guide sequence CC UACCAGGACAUUCAGAAAGCUUU (passenger sequence AAAGCUUUCUGAAUGUCCUGGUAGG ), *Cblb* guide sequence (CUGACUUCUUGGUAUCUGAUAUAUA (passenger sequence UAUAUA UCAGAUACCAAGAAGUCAGGU). Aliquots of 2 M BMDMs in 100 µl opti-MEM (ThermoFisher, Waltham, MA) were transfected with 1 µM siRNA via electroporation at 400 V for 10 ms using a BTX square-wave electroporator. Cells were then plated on 150 mm untreated cell culture dishes and rested in 10 ml DMEM10 for 30 min before adding 10 ml DMEM10 supplemented with sodium pyruvate and 10% CMG-14–12-cell-conditioned medium as described above. Transfections were pooled from several cuvettes to obtain enough cells for several stimulation conditions. After 24 h cells were resuspended in enzyme-free EDTA buffer and replated in untreated 6-well plates (BD Falcon, Sigma-Aldrich) at 1 M cells per well in unconditioned medium. Cells were used 48 hr after transfection.

## saRNA enrichment of c-Cbl

An saRNA sequence designed to upregulate transcription of *Cbl* was designed using an algorithm published previously (*Voutila et al., 2012*). *Cbl*-targeting saRNA 'CBL-Tr-NM_005188-Pr-30-Cp-0' with guide sequence UCAAUUCUAGAUAAAGGCG (passenger sequence CGCCUUUAUCUAGAA UUGA) was ordered from IDT with 2' O-methylated uracil tails (mUmU) to increase transfection efficiency. On the first day of transfection, aliquots of $10^5$ mast cells were resuspended in 400 µl mast-cell medium in a 24-well plate and mixed with 100 µl saRNA + lipofectamine 2000 or 3000 for a final concentration of 5 nM saRNA. Cells were then rested overnight before a second transfection with 100 µl saRNA + lipofectamine. On day three the cells were transferred into fresh mast-cell medium and rested overnight. On the day of the experiment, mast-cell aliquots from the same saRNA condition were pooled, resuspended in 50 µl DMEM, and rested 2 hr before stimulation.

## Cell stimulation and immunoblotting

BMDM stimulations have been described previously (*Freedman et al., 2015*). After resting, 1 M live Jurkat cells, mast cells, or adherent BMDMs were treated at 37°C in DMEM (myeloid cells) or PBS (Jurkat cells) with 10 µM 3-IB-PP1, a gift from K. Shokat (University of California, San Francisco). Signaling reactions were quenched by placing on ice and lysing cells in sodium dodecyl sulfate (SDS) buffer (128 mM Tris base, 10% glycerol, 4% SDS, 50 mM dithiothreitol (DTT), pH 6.8). Whole-cell lysates were prepared for immunoblotting by sonication with a Diagenode Bioruptor (Diagenode, Inc, Denville, NJ) for 3 min and boiling for 15 min. For immunoblotting 0.025 M cell equivalents were run in each lane of a 7% NuPage Tris-Acetate gel (Invitrogen, Carlsbad, CA) and then transferred to an Immobilon-FL PVDF membrane (EMD Millipore, Burlington, MA). REVERT Total Protein Stain (TPS, LI-COR Biosciences, Lincoln, NE) was used according to the standard protocol to quantify lane loading. After destaining, membranes were treated with Odyssey Blocking Buffer (TBS) for at least 1 hr. Blotting was performed using standard procedures, and blots were imaged on an

Odyssey CLx near-infrared imager (LI-COR). Antibodies for immunoblotting were purchased from Cell Signaling Technology (Danvers, MA), ProMab Biotechnologies (Richmond, CA), Thermo Fisher Scientific (Waltham, MA), Abcam (Cambridge, UK), Santa Cruz (Dallas, TX), and LI-COR Biosciences as listed in the key resource table.

## Quantification, statistics, and image processing

Immunoblots were analyzed by densitometry using ImageStudio software (LI-COR). Images were background-subtracted, and bands of the appropriate molecular weight were demarcated and analyzed for each gel lane. Each value was corrected for the total lane protein content (REVERT TPS), with the exception of pErk1/2, which was corrected with reference to total Erk1/2. Data were further normalized to a control condition to show relative changes from a reference state: each-condition t = 0 or reference-condition t = 0 as indicated. For all figures, n values are biological replicates, reflecting independent experiments from different individual mice (except where indicated if sufficient numbers of animals were not available), different days and/or batches of cells, stimulus preparations, and experimental workflows. Statistical analysis was performed using Prism Software (Graphpad, La Jolla, CA). Significance was assessed using one- or two-way ANOVA analysis with Tukey's or Sidak's correction for multiple comparisons, with analysis of the mean at each time point or condition. Unless specified otherwise, error bars represent standard error of the mean from at least three independent experiments. Asterisks reflect specified P-values. Figures were prepared using Graphpad Prism and Adobe Creative Cloud software (San Jose, CA). Where appropriate, images were optimized by applying brightness/contrast changes to the whole image. No gamma or other nonlinear correction was applied. Images were rotated for figure preparation only after densitometry analysis.

## Preparation of macrophage LynA samples for targeted mass spectrometry analysis

BMDMs derived from $Csk^{AS}Cbl^{+/-}$ mice were rested or treated 15 s with 3-IB-PP1, washed with ice-cold PBS, and lysed in 1% Lauryl Maltoside Buffer containing 150 mM NaCl, 0.01% sodium azide, and protease and phosphatase inhibitors (MSSAFE, Sigma-Aldrich, St. Louis, MO). After scraping the plates, cells and detergent were sonicated on a Diagenode Biorupter for 5 min with a 50% duty cycle. The lysate was then cleared by centrifugation for 15 min at 14,000 rpm at 4°C in a tabletop centrifuge. Lysates were precleared for 30 min at 4°C with Protein G Sepharose beads (Sigma-Aldrich) and normal rabbit serum (Jackson ImmunoResearch, West Grove, PA). LynA-specific antibody (*Freedman et al., 2015*) (C13F9, Cell Signaling) was pre-bound to Protein-G-Sepharose beads for at least 2 hr. A bicinchoninic acid (BCA) protein assay was performed on the whole-cell lysates (Thermo Fisher) to ensure that equal amounts of protein from untreated and 3-IB-PP1-treated cells would be subjected to immunoprecipitation. Lysates were then mixed continuously with antibody and beads for 2 hr at 4°C to immunoprecipitate LynA. Samples were applied to micro bio-spin chromatography columns (Bio-Rad, Hercules, CA), washed, and eluted with SDS Sample Buffer containing 125 mM Tris, 10% glycerol, 5% 2-mercaptoethanol, and 25% SDS. Samples were then concentrated to 40 µl using Ultracel-3K centrifugal spin columns (EMD Millipore). Each 40 µl immunoprecipitate sample was then resolved by gel electrophoresis. A bovine serum albumin (BSA) standard curve was used to quantifty the total protein in each IP sample and ensure that equal mass quantities were subjected to trypsin digest. For this calibration, we ran 0.05–20 µg BSA along with the immunoprecipitate samples on a 10% Mini-PROTEAN TGX gel (Bio-Rad) at 170 V. Total protein was visualized without fixation using SimplyBlue Safestain (Thermo Fisher) according to manufacturer's instructions. Following staining, the gel was washed 2 × 1 hr with water and imaged on a LI-COR Odyssey. Staining was quantified by densitometry, and BSA signals vs. mass quantities were fit to a linear function (GraphPad Prism). This function was then used to quantify the total protein in each immunoprecipitate. Gel pieces including lower-MW (nonubiquitinated) Lyn species and higher-MW (polyubiquitinated) species were excised, spiked with the isotope-labeled LynA Y32 phospho-peptide [H]TI[pY]VRDP[$^{13}C_5$$^{15}N_1$]TSNK[OH] (pY32*, Sigma-Aldrich), and subjected to in-gel digestion with TPCK-treated sequencing-grade trypsin (Promega, Madison, WI) and STAGE Tip peptide cleanup as previously described (*Thu et al., 2016*) except that iodoacetamide was used as the

alkylating reagent. Digested samples were submitted for identification at the University of Minnesota Center for Mass Spectrometry and Proteomics.

## Mass spectrometry of LynA immunoprecipitates

Peptide separations were performed on an Easy-nLC 1000 HPLC (ThermoFisher) and loaded directly onto a 75 cm x 100 µm internal diameter fused silica PicoTip Emitter (New Objective, Woburn, MA), packed in-house with ReproSil-Pur C18-AQ (1.9 µm particle, 120 Å pore; Dr. Maish GmbH Ammerbuch, Germany) heated to 55˚C. Peptide elution was performed using a tripartite gradient, decreasing the fraction of Buffer A (0.1% formic acid in water) and increasing the fraction of Buffer B (0.1% formic acid in acetonitrile) at a flow rate of 300 nl/min (Step 1: 5–10% Buffer B over 5 min, Step 2: 10–16% B over 40 min, and Step 3: 16–26% B over 5 min).

The column was mounted in a nanospray source in line with an Orbitrap Fusion mass spectrometer (Thermo Scientific). Spray voltage was 2.1 kV in positive mode, and the heated capillary was maintained at 275˚C. The acquisition method combined two scan events corresponding to a full scan event and a parallel reaction monitoring (PRM) event targeting the singly-, doubly-, and triply-charged precursor ions of the three phosphorylated peptides without scheduling. The full scan event employed a $m/z$ 380–1500 mass selection, an orbitrap resolution of 60,000 (at $m/z$ 200), a target automatic gain control (AGC) value of 4e5, and maximum fill times of 50 ms. The PRM event employed an orbitrap resolution of 30,000 (at $m/z$ 200), a target AGC value of 5e4 with a maximum ion injection time of 54 ms. The precursor ion of each targeted peptide was isolated using an isolation window of 1.6 $m/z$. Fragmentation was performed with a HCD collision energy of 30% and MS/MS scans were collected using a scan range from 100 to 1000 $m/z$. PRM data were collected in centroid mode. Data analysis was performed using manual integration using Thermo Xcalibur Qual Browser and Skyline (*MacLean et al., 2010*). Peak searches were performed using PEAKS Studio X (Bioinformatics Solutions, Waterloo, Ontario). Experiments were run in triplicate, and the entire experiment was performed independently twice.

## Data and materials availability

Reagents and protocols will be provided upon request.

## Acknowledgements

We are indebted to Kevan Shokat and Flora Rutaganira for their synthesis and donation of 3-IB-PP1 for this study. Many thanks also to Arthur Weiss, Clifford Lowell, Clare Abram, Yoji Shimizu, Brandon Burbach, Michael Farrar, Erik Peterson, Juan Rivera, Ryo Suzuki, Marianne Mollenauer, Terri Kadlecek, LeeAnn Higgins, and Whitney Swanson for reagents, methodological expertise, and discussion. Thanks also to Hai-Bin Ruan, John Connett, Marc Jenkins, Bryce Binstadt, Diane Lidke, Nicholas Levinson, Aditi Bapat, Frances Sjaastad, Anders Lindstedt, JT Greene, and Emily Ewan for valuable feedback and discussion of the project or manuscript.

## Additional information

### Funding

| Funder | Grant reference number | Author |
|---|---|---|
| NIH Office of the Director | R01AR073966 | Tanya S Freedman |
| NIH Office of the Director | R03AI130978 | Tanya S Freedman |
| American Cancer Society | UMN IRG-58-001-55 | Tanya S Freedman |
| University of Minnesota | Grant-in-Aid #92286 | Tanya S Freedman |
| University of Minnesota | New Faculty Research Award NF-0315-02 | Tanya S Freedman |
| University of Minnesota | Equipment Award E-0918-01 | Tanya S Freedman |

| University of Minnesota | Center for Autoimmune Diseases Research Pilot Grant | Tanya S Freedman |
|---|---|---|
| NIH Office of the Director | T32DA007097 | Kathryn L Schwertfeger |
| Norwegian Research Council | 230338 | Ben F Brian IV |
| Prostate Cancer Foundation | Young Investigator Award | Pål Sætrom |
| U.S. Department of Defense | Prostate Cancer Research Program W81XWH-18-1-0542 | Justin M Drake |
| NIH Office of the Director | R01CA215052 | Justin M Drake |
| NIH Office of the Director | R01HD095858 | Kathryn L Schwertfeger |

The funders had no role in study design, data collection and interpretation, or the decision to submit the work for publication.

### Author contributions
Ben F Brian IV, Conceptualization, Data curation, Formal analysis, Funding acquisition, Validation, Investigation, Visualization, Methodology, Writing—original draft, Writing—review and editing; Adrienne S Jolicoeur, Formal analysis, Validation, Investigation, Visualization, Writing—original draft, Writing—review and editing; Candace R Guerrero, Formal analysis, Validation, Investigation, Visualization, Methodology, Writing—original draft, Writing—review and editing; Myra G Nunez, Zoi E Sychev, Formal analysis, Validation, Investigation, Visualization, Writing—review and editing; Siv A Hegre, Pål Sætrom, Nagy Habib, Formal analysis, Methodology, Writing—review and editing; Justin M Drake, Formal analysis, Supervision, Methodology, Writing—review and editing; Kathryn L Schwertfeger, Formal analysis, Supervision, Funding acquisition, Project administration, Writing—review and editing; Tanya S Freedman, Conceptualization, Resources, Data curation, Software, Formal analysis, Supervision, Funding acquisition, Validation, Investigation, Visualization, Methodology, Writing—original draft, Project administration, Writing—review and editing

### Author ORCIDs
Pål Sætrom (ID) https://orcid.org/0000-0001-8142-7441
Nagy Habib (ID) https://orcid.org/0000-0003-4920-4154
Tanya S Freedman (ID) https://orcid.org/0000-0001-5168-5829

### Ethics
Animal experimentation: All animal use complies with University of Minnesota (UMN) and National Institutes of Health (NIH) policy (Animal Welfare Assurance Number A3456-01). UMN is accredited by AAALAC, and all animal use was approved by the UMN Institutional Animal Care and Use Committee (IACUC, protocol # 1603-33559A). Animals are kept under supervision of a licensed doctor of veterinary medicine and supporting veterinary staff under strict NIH guidelines.

### Decision letter and Author response
Decision letter https://doi.org/10.7554/eLife.46043.049
Author response https://doi.org/10.7554/eLife.46043.050

## Additional files

### Supplementary files
• Transparent reporting form
DOI: https://doi.org/10.7554/eLife.46043.042

### Data availability
All data generated or analysed during this study are included in the manuscript and supporting files. Source data files have been provided for graphs in Figure 1, Figure 1-figure supplement 1, Figure 2,

Figure 3, Figure 3-figure supplement 2, Figure 4, Figure 4-figure supplement 1, Figure 4-figure supplement 5, Figure 5, Figure 6, Figure 6-figure supplement 1, Figure 7, Figure 8, and Figure 9. Data sets and calibration curves resulting from our targeted mass spectrometry studies have been deposited in Panorama Public (https://panoramaweb.org/project/Panorama%20Public/begin.view?).

The following dataset was generated:

| Author(s) | Year | Dataset title | Dataset URL | Database and Identifier |
|---|---|---|---|---|
| Freedman T | 2019 | Unique-region phosphorylation targets LynA for rapid degradation, tuning its expression and signaling in myeloid cells | http://panoramaweb.org/Freedman_LynA.url | Panorama, Freedman_LynA |

The following previously published datasets were used:

| Author(s) | Year | Dataset title | Dataset URL | Database and Identifier |
|---|---|---|---|---|
| Heng TS, Painter MW | 2016 | Immunological Genome Project C. Expression profiling of constitutive mast cells reveals a unique identity within the immune system | https://www.ncbi.nlm.nih.gov/geo/query/acc.cgi?acc=GSE37448 | NCBI Gene Expression Omnibus, GSE37448 |

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
