## [Decision Letter]

Thank you for submitting your article "An orthogonal c-Cbl recognition mode targets LynA for rapid degradation and builds specificity into the LynA checkpoint" for consideration by *eLife*. Your article has been reviewed by three peer reviewers, one of whom is a member of our Board of Reviewing Editors, and the evaluation has been overseen by Tadatsugu Taniguchi as the Senior Editor. The reviewers have opted to remain anonymous.

The reviewers have discussed the reviews with one another and the Reviewing Editor has drafted this decision to help you prepare a revised submission.

Summary:

This is an interesting study that builds upon previous work by these authors that demonstrates a possible check-point role for LynA in macrophage activation caused by activation-induced degradation. This study demonstrates that this degradation is mediated by the E3 Ub ligase cCbl that targets the insert region on LynA and requires Tyr32. The data supporting the authors' conclusions are strong.

The major weakness of the study is that cCbl has already been reported to cause Lyn degradation. Nevertheless, the current study has merit because it demonstrates that this role is specific for LynA and because the study presents these observations within an important biological context. Freedman and co-workers demonstrate a rapid negative feedback loop in a specific spliced form of the Src family tyrosine kinase LynA that is not seen with the alternatively spliced form, LynB, or with other members of the SFK family. This is physiologically significant as MAST cells that use LynB, but not LynA, do not exhibit this phenotype and have a more sustained and higher levels of ERK activation.

Essential revisions:

1) What is the meaning of "orthogonal" in the title?

2) What is the function of Y32? Why does its phosphorylation enhance ubiquitination and degradation? It is suggested that this could be a binding site for cCBL as well as an activating phosphorylation? Was the binding of cCBL tested? This would be a logical mechanism of action. Does Y32 phosphorylation induce relocalization of Lyn into the cytoplasm from the membrane? That could also explain the mechanism.

3) Does the Y32 get autophosphorylated in an in vitro kinase reaction? If so, is this phosphorylation in cis or trans? For example, is autophosphorylation concentration-dependent in vitro. Can the Y32 phosphorylation be reconstituted using kinase-inactive and phosphorylation site mutated proteins?

4) What happens in mast cells if c-CBL is expressed? Would it result in loss of resting LynA? Would that attenuate ERK activation?

---

## [Author Response]

Essential revisions:1) What is the meaning of "orthogonal" in the title?

pY32-dependent recognition of LynA by c-Cbl is distinct from the known PXXP/SH3, PTB/pSFK activation loop, and kinase/substrate interactions mediating the slower degradation of all SFKs. We now convey this point in a less oblique manner in the revised title “Unique-region phosphorylation targets LynA for rapid degradation, tuning its expression and signaling in myeloid cells.” We have also revised the abstract and main text to emphasize the difference between canonical c-Cbl/SFK interactions and the preferential targeting of LynA for rapid polyubiquitination by c-Cbl.

2) What is the function of Y32? Why does its phosphorylation enhance ubiquitination and degradation?

In the original manuscript we reported that unique-region Y32 is the key residue marking LynA for rapid polyubiquitination and degradation. We speculated that this might be a site of phosphorylation, based on a prediction algorithm (NetPhos 2.0) and recent reports of this modification in tumor cells. However, we had no evidence that this site was actually phosphorylated in non-cancer cells or in hematopoietic cells, and thus the mechanism of Y32 in targeting LynA for rapid polyubiquitination was unclear. We have now performed quantitative, targeted LC-MS/MS analysis of LynA immunoprecipitates from resting and 3-IB-PP1-treated macrophages. The results of these experiments are reported in new Figure 4 and Figure 4—figure supplements 1-5. We can now unequivocally state that in primary macrophages LynA Y32 is a site of phosphorylation, induced upon SFK activation. Moreover, pY32 is massively enriched in polyubiquitinated LynA. We therefore conclude that activation of the SFKs leads to phosphorylation of LynA at position Y32, and that this is the mechanism by which LynA is targeted specifically for polyubiquitination and degradation.

It is suggested that this could be a binding site for cCBL as well as an activating phosphorylation? Was the binding of cCBL tested? This would be a logical mechanism of action.

These mass spectrometry experiments were quite labor intensive and occupied the entire revision time frame. We did perform co-immunoprecipitation experiments and found inducible association of c-Cbl with LynA, but we were unable to demonstrate that this association was due to Y32 phosphorylation. Like all SFKs, LynA interacts with c-Cbl via several binding interfaces, including the SH3 domain, the activation-loop peptide binding site, and the activation-loop phosphotyrosine. Additional cloning to isolate the contributions of each of these interfaces was not possible before resubmission.

Does Y32 phosphorylation induce relocalization of Lyn into the cytoplasm from the membrane? That could also explain the mechanism.

We probed for localization differences mediated by the unique regions of LynA, LynB, and LynA^Y32A^ using epifluorescence microscopy. We were unable to get good staining of total LynA, but we were able to visualize eGFP-tagged unique-region constructs. As shown in new Figure 3—figure supplement 1. All three constructs were membrane localized in the presence and absence of 3-IB-PP1, even in the presence of a kinase-active LynB construct. While the LynA kinase domain may add some localization functionality, we can say that the Y32 mutation does not itself change the localization of the LynA unique region.

3) Does the Y32 get autophosphorylated in an in vitro kinase reaction? If so, is this phosphorylation in cis or trans? For example, is autophosphorylation concentration-dependent in vitro.

We used targeted mass spectrometry to detect phosphorylation of LynA Y32 with increasing doses of recombinant LynA (to probe cis vs. transautophosphorylation) or increasing doses of recombinant Hck (to probe transphosphorylation). Unfortunately, the data were uninterpretable because recombinant LynA purified from insect cells is already phosphorylated at position Y32. We are continuing to troubleshoot these experiments, but they are likely to take several more months and so unfortunately were not possible for this revision.

Can the Y32 phosphorylation be reconstituted using kinase-inactive and phosphorylation site mutated proteins?

We measured the ability of different Src family members to induce degradation of kinase-dead LynA^T410K^, which is not on its own degraded during 3-IB-PP1 treatment. LynA degradation can be specifically induced in trans by LynA, LynB, LynA^Y32A^, and Hck^56^ (the shorter isoform of Hck). In contrast, Fyn, Fgr, Hck^59^, and Lck were unable to induce degradation of LynA. These data, presented in new Figure 7 and Figure 7—figure supplement 1-2, reveal SFK-specific regulation of LynA degradation, with LynA itself likely inducing its own degradation via Y32 autophosphorylation.

4) What happens in mast cells if c-CBL is expressed? Would it result in loss of resting LynA? Would that attenuate ERK activation?

We treated mast cells with small activating (sa)RNAs designed to increase transcription of c-Cbl mRNA. One of these constructs was found to increase c-Cbl expression an average of 2-fold, and we were able to detect an increase in LynA degradation and corresponding decrease in Syk and Erk phosphorylation in response to 3-IB-PP1 in c-Cbl overexpressing mast cells relative to mock-transfected cells (new panels, Figure 9G-J). With this modest increase in c-Cbl expression, steady-state expression of LynA was not significantly affected.